# PI(18:1/18:1) is a SCD1-derived lipokine that limits stress signaling

Maria Thürmer[1], André Gollowitzer[2,16], Helmut Pein[1,16], Konstantin Neukirch[2,16], Elif Gelmez[1], Lorenz Waltl[2], Natalie Wielsch[3], René Winkler[4], Konstantin Löser[1], Julia Grander[2], Madlen Hotze[5], Sönke Harder[6], Annika Döding[7], Martina Meßner[8,9], Fabiana Troisi[1], Maximilian Ardelt[8,9], Hartmut Schlüter[6], Johanna Pachmayr[8,9], Óscar Gutiérrez-Gutiérrez[10], Karl Lenhard Rudolph[11], Kathrin Thedieck[5,12,13], Ulrike Schulze-Späte[7], Cristina González-Estévez[10,14,15], Christian Kosan[4], Aleš Svatoš[3], Marcel Kwiatkowski[5] & Andreas Koeberle[1,2✉]

Cytotoxic stress activates stress-activated kinases, initiates adaptive mechanisms, including the unfolded protein response (UPR) and autophagy, and induces programmed cell death. Fatty acid unsaturation, controlled by stearoyl-CoA desaturase (SCD)1, prevents cytotoxic stress but the mechanisms are diffuse. Here, we show that 1,2-dioleoyl-*sn*-glycero-3-phospho-(1'-myo-inositol) [PI(18:1/18:1)] is a SCD1-derived signaling lipid, which inhibits p38 mitogen-activated protein kinase activation, counteracts UPR, endoplasmic reticulum-associated protein degradation, and apoptosis, regulates autophagy, and maintains cell morphology and proliferation. SCD1 expression and the cellular PI(18:1/18:1) proportion decrease during the onset of cell death, thereby repressing protein phosphatase 2 A and enhancing stress signaling. This counter-regulation applies to mechanistically diverse death-inducing conditions and is found in multiple human and mouse cell lines and tissues of *Scd1*-defective mice. PI(18:1/18:1) ratios reflect stress tolerance in tumorigenesis, chemoresistance, infection, high-fat diet, and immune aging. Together, PI(18:1/18:1) is a lipokine that links fatty acid unsaturation with stress responses, and its depletion evokes stress signaling.

---

[1] Chair of Pharmaceutical/Medicinal Chemistry, Institute of Pharmacy, Friedrich-Schiller-University Jena, 07743 Jena, Germany. [2] Michael Popp Institute and Center for Molecular Biosciences Innsbruck (CMBI), University of Innsbruck, 6020 Innsbruck, Austria. [3] Research Group Mass Spectrometry and Proteomics, Max Planck Institute for Chemical Ecology, 07745 Jena, Germany. [4] Department of Biochemistry, Center for Molecular Biomedicine (CMB), Friedrich-Schiller-University Jena, 07745 Jena, Germany. [5] Institute of Biochemistry and Center for Molecular Biosciences Innsbruck, University of Innsbruck, 6020 Innsbruck, Austria. [6] Institute of Clinical Chemistry and Laboratory Medicine, Section Mass Spectrometry and Proteomics, University Medical Center Hamburg-Eppendorf, 20246 Hamburg, Germany. [7] Section of Geriatric Dentistry, Center of Dental Medicine, University Hospital Jena, Friedrich-Schiller-University Jena, 07743 Jena, Germany. [8] Department of Pharmacy, Pharmaceutical Biology, LMU Munich, 81377 Munich, Germany. [9] Institute of Pharmacy, Paracelsus Medical University, 5020 Salzburg, Austria. [10] Leibniz Institute on Aging—Fritz Lipmann Institute (FLI), 07745 Jena, Germany. [11] Research Group on Stem Cell Aging, Leibniz Institute on Aging—Fritz Lipmann Institute (FLI), 07745 Jena, Germany. [12] Laboratory of Pediatrics, Section Systems Medicine of Metabolism and Signaling, University of Groningen, University Medical Center Groningen, 9713 AV Groningen, The Netherlands. [13] Department of Neuroscience, School of Medicine and Health Sciences, Carl von Ossietzky University Oldenburg, 26129 Oldenburg, Germany. [14] Department of Genetics, Microbiology and Statistics, Faculty of Biology, University of Barcelona, 08028 Barcelona, Spain. [15] Institute of Biomedicine of the University of Barcelona (IBUB), 08028 Barcelona, Spain. [16]These authors contributed equally: André Gollowitzer, Helmut Pein, Konstantin Neukirch. ✉email: andreas.koeberle@uibk.ac.at

Fatty acid unsaturation links cell metabolism with stress signaling[1,2]. Excess saturated fatty acids (SFAs) cause lipotoxic stress, whereas polyunsaturated fatty acids (PUFAs) render membranes more susceptible to oxidative damage[3,4]. Cells have developed manifold strategies to sense stress and adapt to metabolic challenges during evolution, which include stress-activated protein kinases[5,6], the unfolded protein response (UPR)[7], and autophagy[8].

Stress-activated protein kinases play an important role in inflammation and cell homeostasis, as they regulate proliferation, survival, metabolism, and differentiation[5]. By participating in cytotoxic stress signaling and stress adaption, they either promote persistence or initiate programmed cell death[5,6]. The stress-activated p38 mitogen-activated protein kinase α (MAPK14, p38 MAPK) is ubiquitously expressed and activated within a sequential kinase cascade[9–11]. Upon phosphorylation of Thr[180] and Tyr[182] in the activation loop, p38 MAPK phosphorylates a myriad of downstream substrates, including transcription factors, mitogenic kinases, and pro-apoptotic factors that mediate stress responses but are also implicated in processes not related to stress[5,6,10,12]. Among others, p38 MAPK facilitates the induction and progression of apoptosis[6], induces the UPR[3,13], inhibits autophagy[14,15], couples endoplasmic reticulum (ER) stress to chaperone-mediated autophagy[16,17], and contributes to tumor survival and resistance[5,18]. p38 MAPK is activated by genotoxic, inflammatory, and metabolic stress[5,10], such as high concentrations of SFAs, which induce lipotoxic ER stress at physiologically relevant concentrations[3].

Stress-protective mechanisms like the UPR or autophagy, which are activated in parallel, either succeed in maintaining organelle function or initiate programmed cell death. They induce intrinsic apoptosis[19,20] or, in case of selective autophagy, additionally promote ferroptosis[21], a recently described necrotic programmed cell death pathway based on lipid peroxidation[4,22,23]. Cytosolic components are degraded in autophagy and the breakdown products are recycled to supply the energy to maintain stress-protective mechanisms[20], e.g., the conversion of excess SFAs to monounsaturated fatty acids (MUFAs)[24,25]. MUFAs are less efficient than SFAs in inducing stress(-adaptive) responses or even counteract SFA-triggered effects[3,26].

The SFA/MUFA ratio is influenced by systemic parameters, such as the diet, and adjusted within the cell by ubiquitously expressed stearoyl-CoA desaturases (SCDs) that introduce a Δ9-cis-double bond into SFA-coenzyme A (CoA)[27]. Inhibition of the isoenzyme SCD1 evokes a shift from MUFAs as major cellular fatty acids towards SFAs and PUFAs throughout cellular lipids[3,24,28,29]. In consequence, ER stress and apoptosis are induced, the susceptibility to ferroptosis enhanced, and stress-adaptive responses initiated, including the p38 MAPK cascade, the UPR, and autophagy[3,24,30,31]. SCD1 is explored as pharmacological target in metabolic diseases, skin disorders and cancer, and selective inhibitors of SCD1 are currently under clinical investigation[32,33]. While plenty of studies on cells, animals, and humans describe different biological functions of SFAs and MUFAs, the understanding of the metabolites and physiologically relevant molecular mechanisms by which fatty acid unsaturation regulates stress signaling is fragmentary[3]. Mechanisms discussed focus on specific receptors[34], membrane anchors[35], redox properties[30], and changes in membrane rigidity, fluidity, permeability, or microdomain structure[36,37]. Several studies speculated about a role of SCD1 in the biosynthesis of MUFA-derived bioactive lipids[3].

Here, we report on the identification of 1,2-dioleoyl-*sn*-glycero-3-phospho-(1'-myo-inositol) [PI(18:1/18:1)] as SCD1-derived lipokine that promotes cell survival and counteracts cellular stress responses by interfering with stress-activated pathways, i.e., p38 MAPK signaling, the UPR, and autophagy. The drop of PI(18:1/18:1) levels during the onset of programmed cell death enhances p38 MAPK stress signaling across cytotoxic conditions and cell lines and is associated with tissue-specific stress responses in *Scd1*-defective mice. Quantitative proteomics highlights the catalytic subunit of protein phosphatase 2 A (Ppp2ca) as SCD1/PI(18:1/18:1)-regulated protein that depletes during cytotoxic cell stress and participates in p38 MAPK[38–40], UPR[41], and autophagy regulation[42]. We further show that PI(18:1/18:1) levels are responsive to physiological stress conditions, including tumorigenesis, chemoresistance, infection, dietary restriction, and aging, and outline exemplary links to stress(-adaptive) signaling.

## Results

**MUFAs in PI deplete during programmed cell death**. We induced programmed cell death in fibroblasts through conditions that cover a broad mechanistic range. Cell death was triggered by (i) pan-kinase inhibition (staurosporine, STS)[43], (ii) the blockage of protein biosynthesis (cycloheximide, CHX)[44], (iii) topoisomerase inactivation that leads to DNA strand breaks (etoposide, ETO)[45], (iv) the disruption of K[+] gradients (valinomycin, VAL)[46], (v) ER stress induction by depletion of ER Ca[2+] stores (thapsigargin, TPG)[47], (vi) interference with mitochondrial function by targeting heat-shock protein 60 (myrtucommulone A, MC)[48], (vii) cell cycle arrest upon inhibition of cyclin-dependent kinases and glycogen synthase kinase-3β (indirubin-3'-monoxime, I3M)[49,50], and (viii) the withdrawal of nutrients and growth factors by serum depletion. Moreover, fibroblasts were sensitized to cytotoxic stress by tumor necrosis factor (TNF)α without inducing apoptosis per se[51].

We monitored the phospholipid composition of still attached (viable) fibroblasts under these cytotoxic conditions over 48 h and combined the data in a co-regulated phospholipid network (Supplementary Fig. 1a). Positively correlated phospholipids are located in close proximity and interconnected, whereas non- or counter-regulated phospholipids form separate clusters. Our focus was placed on the lower left cluster, which is dominated by phosphatidylcholines (PC), phosphatidylethanolamines (PE), phosphatidylserines (PS), and PI species that contain one or two MUFAs (Supplementary Fig. 1a).

The cellular proportion of phospholipid species containing MUFAs, i.e., palmitoleic acid (16:1) or oleic acid (18:1), was substantially decreased by diverse cytotoxic settings (i.e., STS, CHX, ETO, TPG, VAL, serum depletion, and marginally MC) after 6–48 h (Supplementary Fig. 1b), with strongest effects on PIs (Fig. 1a). Kinetic data on total PI and MUFA-PI levels are shown in Fig. 1b and Supplementary Fig. 1c. PIs containing two MUFAs were even more consistently down-regulated in programmed cell death than species that combine 18:1 with either SFAs or PUFAs. Thus, PI(16:1/18:1) and PI(18:1/18:1) were markedly reduced for all cytotoxic stimuli studied (Fig. 1a, c, Supplementary Fig. 1d), including I3M and MC, which failed to decrease the MUFA ratio in PI (Fig. 1a). MUFAs in other phospholipid classes were less affected, and only CHX, TPG, and VAL substantially lowered the MUFA ratio of PC, PE, or PS (Fig. 1d–f and Supplementary Fig. 1e, f).

The drop in MUFA-containing PI species (MUFA-PIs) was accompanied by an increased proportion of species with PUFAs, i.e., eicosatrienoic acid (20:3), arachidonic acid (20:4), docosapentaenoic acid (22:5), and docosahexaenoic acid (22:6) (Fig. 1c). Since total PI levels are not substantially upregulated under cytotoxic conditions (Fig. 1b), the relative enrichment of PUFA-containing PIs reflects an increase in absolute numbers.

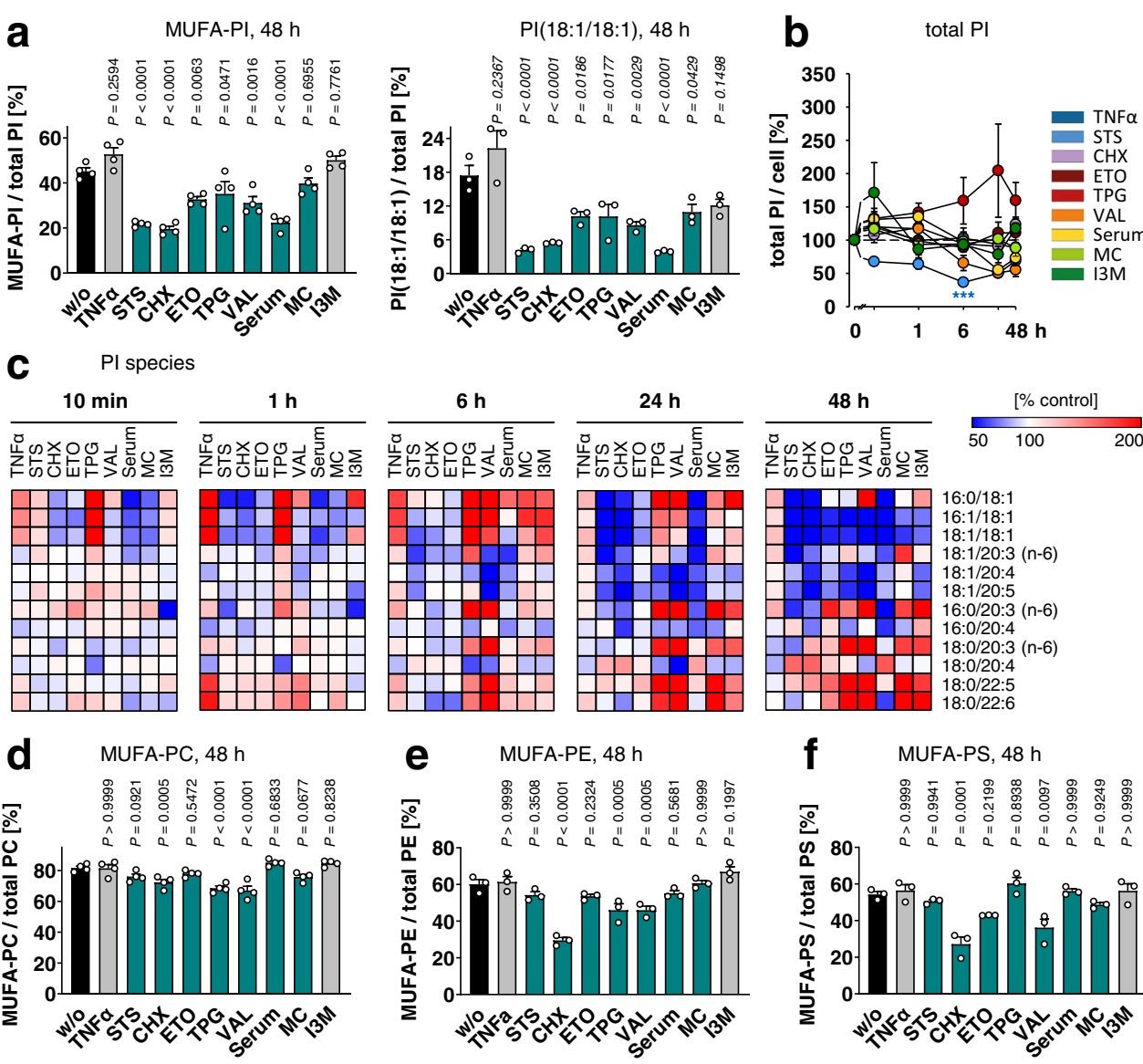

**Fig. 1 Programmed cell death decreases the cellular proportion of MUFAs in PI.** NIH-3T3 fibroblasts were treated with vehicle, TNFα (10 ng/ml), STS (0.3 μM), CHX (20 μg/ml), ETO (10 μM), TPG (2 μM), VAL (10 μM), MC (10 μM), or I3M (10 μM) or were serum starved (Serum) for the indicated period of time. **a**, **d–f** Cellular proportion of MUFAs in PI and PI(18:1/18:1) (left to right (LTR) $P = 0.0000004$, $0.00000007$, $0.0000007$) (**a**), PC (LTR $P = 0.9999999986$, $0.000002$, $0.0000003$) (**d**), PE (LTR $P = 0.99991$, $0.000000007$, $0.99999992$) (**e**), and PS (LTR $P = 0.99992$, $0.99998$, $0.99997$) (**f**); MUFAs: 16:1, 18:1. Data of (**a**), (**d**) is identical to w/o in Supplementary Figs. 13a and 16c, d. **b** Time-dependent changes of the cellular PI content. **c** Heatmap showing the time-dependent changes of the cellular PI profile ($P = 0.0007$). Data are given as percentage of vehicle control for each time point. Mean (**c**) or mean + s.e.m. (**b**) and single data (**a**, **d–f**) from $n = 3$ (**a** right panel, **b**, **c**, **e**, **f**), $n = 4$ (**a** left panel, **d**) independent experiments. ***$P < 0.001$ for the respective time point (**b**) or $P$ values given vs. vehicle control (**a**, **d–f**); repeated measures one-way ANOVA (**a**, **d–f**) of log data (**b**) + Tukey HSD post hoc tests.

Lipidomic analysis suggests that PUFAs and MUFAs are redistributed during the initiation of cell death. Thus, PI(palmitic acid (16:0)/18:1), PI(16:1/18:1), and PI(18:1/18:1) are enriched for multiple cytotoxic stressors 6 h post cell death induction before the depletion of MUFAs becomes dominating (Fig. 1c).

**Cytotoxic drop in MUFA-PI correlates with active p38 MAPK.** We investigated whether death-induced changes in the phospholipid profile are associated with the regulation of stress-activated kinases. In particular, phospholipids from the MUFA-rich cluster showed a negative correlation to p38 MAPK phosphorylation (Fig. 2a) in line with previous studies that addressed ER stress and cell cycle M/G1 transition[29]. p38 MAPK was rapidly activated within 10 min to 6 h

and then experienced an even stronger boost in activation up to 48 h (Fig. 2b, c). This second phase of p38 MAPK phosphorylation has similar kinetics to the decrease of the cellular MUFA-PI ratios (Fig. 1c and Supplementary Fig. 1b, c). Both effects manifested between 6 to 48 h of treatment and were time-dependently enhanced. Since STS is a pan-kinase inhibitor[43], we did not further consider its effect on kinase phosphorylation, although p-p38 MAPK levels were elevated as expected. Substantial activation of JNK, another major stress-activated kinase, was only evident for TPG and MC (Fig. 2d), which suggests that the global negative correlation of MUFA-PI is p38 MAPK specific.

To further investigate whether MUFA-PI ratios and p38 MAPK activation correlate during cell death in other cell lines, we

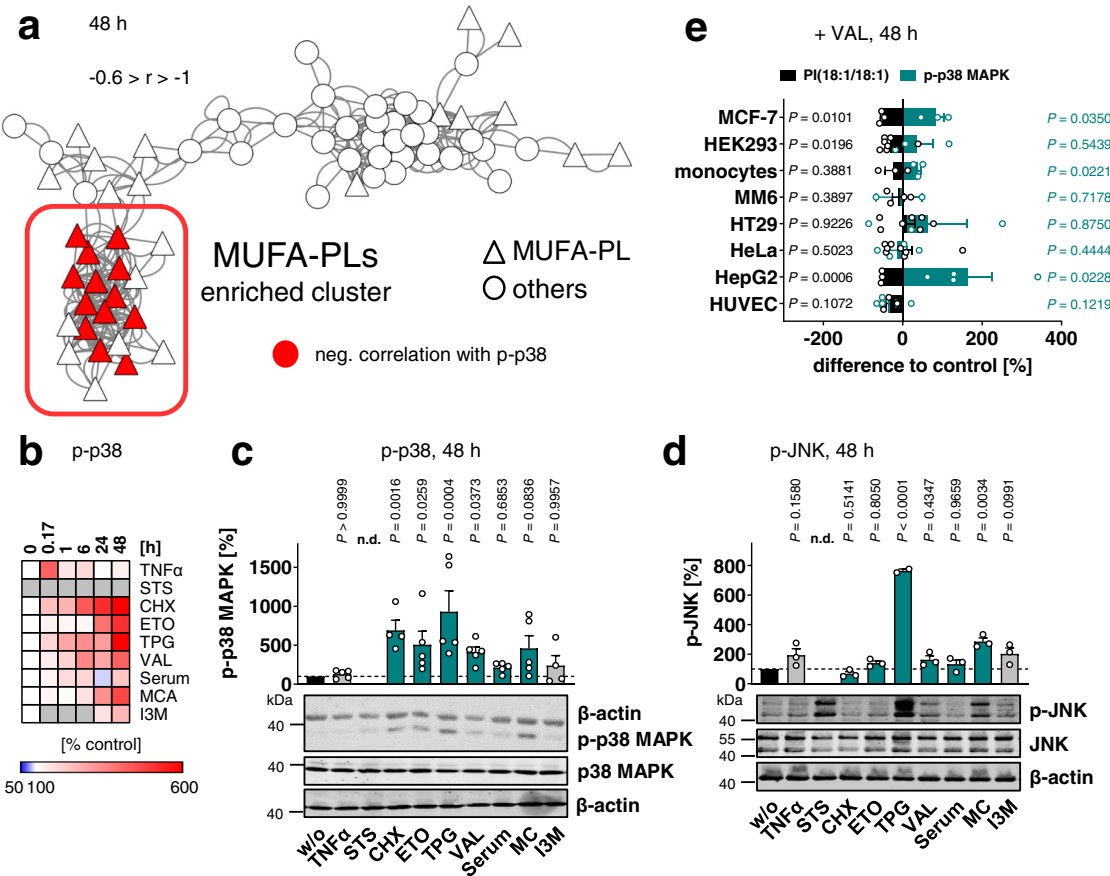

**Fig. 2 p38 MAPK activation accompanies the early cytotoxic decrease of MUFA-PI.** Fibroblasts were cultivated under diverse cytotoxic conditions for 48 h or as indicated. **a** Negative correlation ($-0.6 > r > -1$) between cellular p-p38 MAPK (Thr180/Tyr182) levels (at 48 h) and the proportions of phospholipid (PL) species are shown for the co-regulated lipid network described in Supplementary Fig. 1a. Correlations were calculated for mean p-p38 MAPK levels from three independent experiments. **b** Heatmap showing time-dependent changes in the activation of p38 MAPK compared to vehicle control for each time point. Representative Western blots are shown in Supplementary Fig. 2a. STS, excluded due to pan-kinase inhibition; gray color for I3M, not determined. **c**, **d** Phosphorylation and expression of p38 MAPK ($P = 0.99995$) (**c**) and JNK ($P = 0.000007$) (**d**). Western blots are representative of five (**c**) or three (**b**, **d**) independent experiments. Data of **c** is identical to w/o in Supplementary Fig. 13b, 16b and 25a. Mean (**b**) or mean + s.e.m. and single data (**c**, **d**) from $n = 1$ (**b** I3M for 24 h), $n = 2$ (**b** TNFα for 0.17 h, **d** for TPG), $n = 3$ (**a**, **b**, **d**), $n = 4$ (**b**, **c** for CHX, I3M at 48 h), $n = 5$ (**b** for 48 h, **c**) independent experiments. $P$ values given vs. vehicle control; mixed-effects model (REML) + Tukey HSD post hoc tests of log data (**c**, **d**). **e** Counter-regulation of PI(18:1/18:1) ratios and p38 MAPK activation during VAL-induced cell death across cell lines. MCF-7 breast adenocarcinoma cells, HEK293 embryonic kidney cells, primary human monocytes, MM6 acute monocytic leukemia cells, HT29 colon adenocarcinoma cells, HeLa cervical carcinoma cells, HepG2 hepatoma cells, and HUVECs were treated with vehicle or VAL (10 μM) for 48 h. Percentage changes in cellular PI(18:1/18:1) ratios and p-p38 MAPK levels were calculated vs. vehicle (100%), and the difference to the vehicle control is presented. Representative Western blots are shown in Supplementary Fig. 2b. Detailed descriptions of datasets shown in panel **e** are given in Supplementary Note 2. $P$ values given vs. vehicle control; two-tailed paired student $t$ test.

selected VAL, which caused a representative, average decrease of MUFA-PI levels in NIH-3T3 fibroblasts (Fig. 1a). The negative co-regulation of PI(18:1/18:1) and p38 MAPK is not limited to apoptotic fibroblasts, but was also found in human MCF-7 breast cancer and human HepG2 hepatocarcinoma cells (Fig. 2e). Moreover, we observed trends to lower PI(18:1/18:1) ratios and elevated p38 MAPK activation for VAL-treated human HEK-293 embryonic kidney cells and primary human monocytes. On the other hand, VAL neither substantially decreased the proportion of PI(18:1/18:1) nor enhanced p38 MAPK phosphorylation in human MM6 monocytic cells, human HT-29 colon adenocarcinoma cells, and human HeLa cervix carcinoma cells, and both parameters were reduced in human umbilical vein endothelial cells (HUVECs) (Fig. 2e). This heterogeneity is not surprising in light of the experimental design (Supplementary Note 1) and the variable connectivity of the p38 MAPK signaling network for different cell types[5,6,9,10]. Together, the cellular proportion of PI(18:1/18:1) decreases in (pre)apoptotic cells for various

cytotoxic mechanisms, and the depletion of this lipid is accompanied by the induction of p38 MAPK stress signaling across diverse cell lines.

**MUFA-depletion and cytotoxic stress due to SCD1 repression.** To elucidate how cell death lowers MUFA-PI levels, we first investigated whether the availability of non-esterified MUFAs and lyso-PI (LPI) is affected by cytotoxic stress in fibroblasts. Principal component analysis shows that the MUFAs 16:1 and 18:1, located in the lower left quadrant, are separately regulated from the bulk of fatty acids that are clustered in the lower right quadrant (Supplementary Fig. 3a). The proportion of free MUFAs markedly decreased throughout the cytotoxic settings, whereas the ratio of SFAs increased (Fig. 3a–c, Supplementary Figs. 3b), except for I3M, which neither substantially lowered the proportion of free (Fig. 3a) nor PI-bound MUFAs (Fig. 1a). Lyso PI (LPI) species (16:0-LPI, 18:0-LPI, 18:1-LPI) were instead

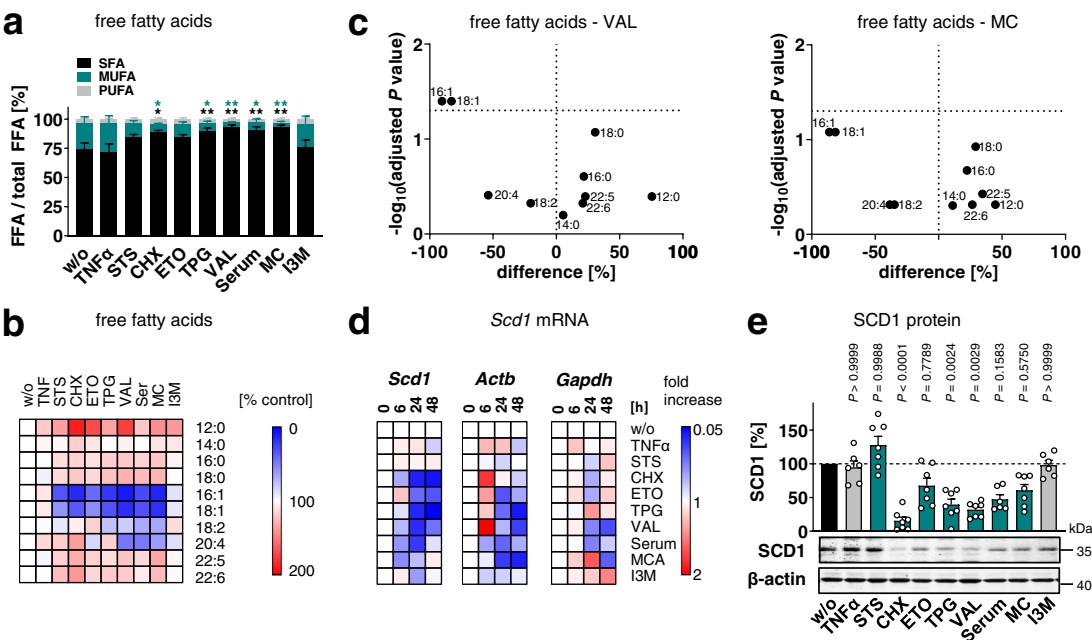

**Fig. 3 SCD1 expression decreases during preparation for cell death.** Fibroblasts were cultivated under diverse cytotoxic conditions for 48 h (**a**–**c**, **e**) or as indicated (**d**). **a** Cellular proportion of non-esterified SFAs, MUFAs, and PUFAs. SFAs: 12:0, 14:0, 16:0, 18:0; MUFAs: 16:1, 18:1; PUFAs: 18:2, 20:4, 22:5, 22:6 (MUFA LTR $P = 0.0146, 0.0128, 0.0023, 0.0155, 0.003$; SFA LTR $P = 0.0176, 0.0088, 0.0013, 0.0072, 0.0015$). **b** Heatmap showing changes in the free fatty acid profile as compared to vehicle control. Data are given as percentage of the relative free fatty acid abundance. **c** Volcano plots highlighting free fatty acids that are strongly and significantly modulated by VAL or MC. Comparisons of the indicated treatment groups show the mean difference of percentage changes and the negative log10(adjusted $P$ value). Adjusted $P$ values given vs. vehicle control; two-tailed multiple unpaired student $t$ tests from log data with correction for multiple comparisons using a two-stage linear step-up procedure by Benjamini, Krieger, and Yekutieli (false discovery rate 5%). **d** Heatmaps showing the time-dependent effect on $Scd1$, $Actb$, and $Gapdh$ mRNA levels that were normalized to the total amount of cellular RNA and compared to vehicle control for each time point. **e** Protein expression of SCD1. Western blots are representative of seven independent experiments (LTR $P = 0.000000002, 0.99$). Mean (**b**–**d**) or mean + s.e.m. (**a**) and single data (**e**) from $n = 2$ (**d** for $Scd1$ and $Gapdh$ at 6 h; $Actb$ at 48 h for Serum), $n = 3$ (**a**–**d**), $n = 6$ (**e** for TNFα, Serum, I3M), $n = 7$ (**e**) independent experiments. *$P < 0.05$, **$P < 0.01$ or $P$ values given vs. vehicle control; repeated measures one-way ANOVA (**a**) or mixed-effects model (REML) of log data (**e**) + Tukey HSD post hoc tests.

differentially regulated under the four cytotoxic settings investigated (Supplementary Fig. 3c). While TPG and serum depletion and, by trend, MC increased the proportion of distinct LPI species, VAL did not affect cellular LPI ratios, and neither of the cytotoxic stressors showed a preference for 18:1-LPI, the phospholipase A$_2$ (PLA$_2$) cleavage product of PI(18:1/18:1). Our data thus indicates that MUFA biosynthesis is diminished by cytotoxic stress and rather excludes (MUFA-selective) PI degradation by phospholipases as dominating mechanism for the depletion of PI(18:1/18:1).

The de novo biosynthesis of SFAs and MUFAs depends on the concerted action of acetyl-CoA carboxylase (ACC) and fatty acid synthase (FAS). Selective inhibition of ACC with soraphen A or siRNA neither decreased the proportion of phospholipid-bound MUFAs nor induced p38 MAPK signaling (Supplementary Fig. 4 and Supplementary Note 3).

The balance between SFAs and MUFAs is adjusted by Δ9-desaturases, with SCD1 being subject to intensive transcriptional regulation[3,24]. In fact, many cytotoxic agents substantially decreased $Scd1$ mRNA levels between 6 to 48 h (Fig. 3d and Supplementary Fig. 5) and SCD1 protein expression at 48 h (Fig. 3e). Exceptions are I3M, which also failed to reduce free MUFA (Fig. 3a) and MUFA-PI ratios (Fig. 1a), and STS, which decreases both free (Fig. 3a) and esterified MUFAs (Fig. 1a) via a SCD1-independent mechanism. Since MUFAs are produced by SCD1 and incorporated into phospholipids as CoA-esters[3], we were surprised to find MUFA-CoA levels being maintained during the initiation of programmed cell death (Supplementary Fig. 6), which suggests that MUFAs from sources other than

SCD1 compensate for the cytotoxic loss of MUFA-CoAs and are poorly channeled into PI biosynthesis.

The role of SCD1 in fibroblast homeostasis and stress signaling was investigated using the selective SCD1 inhibitor CAY10566 and by transient knockdown. CAY10566 (i) enhanced p38 MAPK phosphorylation (Fig. 4a) in confirmation of our previous study[29], (ii) shifted the acyl-CoA ratio from MUFAs to SFAs (Fig. 4b), and (iii) decreased the cellular proportion of MUFA-PI and PI(18:1/18:1) rather than MUFA-PC (Fig. 4c, Supplementary Fig. 7, Supplementary Fig. 8a) without substantially reducing the absolute amount of PI (Supplementary Fig. 8b). Comparable effects were observed when $Scd1$ was silenced by siRNA (Fig. 4d, e). Knockdown efficiencies of siRNAs at mRNA and protein levels are shown in Supplementary Fig. 9a, b.

Major kinases that activate p38 MAPK are the MAPK kinase (MKK)3 and MKK6 and less MKK4[5,6]. SCD1 inhibition by CAY10566 induced MKK3/6 phosphorylation with comparable kinetics to p38 MAPK (Supplementary Fig. 10a, b), whereas MKK4 was not activated (Supplementary Fig. 10c). Next, we investigated putative MAPK kinases kinases (MAP3K) that might phosphorylate MKK3/6 and identified mixed lineage kinase (MLK)3 to be activated upon treatment with CAY10566 (Supplementary Fig. 10d).

Markers of ER stress (binding protein, BiP) (Fig. 4f and Supplementary Fig. 11a), autophagy (light chain (LC)3B II) (Fig. 4g), and apoptosis (cleaved PARP, Fig. 4h; PS externalization, Fig. 4i and Supplementary Fig. 11b) were substantially upregulated in CAY10566-treated fibroblasts, as expected from $Scd1$ knockout studies[3,24]. Fibroblasts acquired a stretched, spindle-shaped

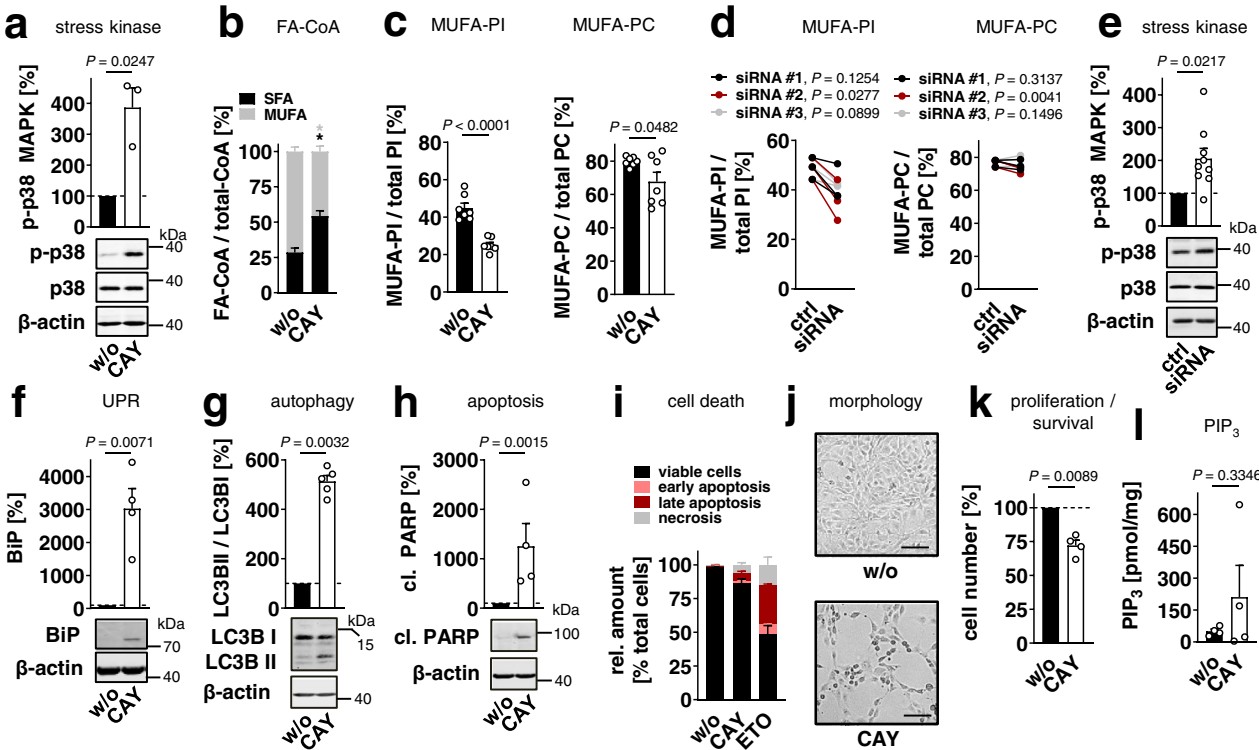

**Fig. 4 SCD1 inhibition lowers MUFA-PI levels and induces stress signaling.** Fibroblasts were treated with CAY10566 (CAY, 3 μM) (**a–c**, **f–l**), ETO (10 μM) (**i**) or *Scd1* siRNA (**d**, **e**) for 48 h. **a**, **e** Phosphorylation of p38 MAPK. Western blots are representative of three independent experiments. Data for **e** is identical to Fig. 6h. **b** Cellular proportion of SFA-CoAs and MUFA-CoAs. SFA: 16:0-CoA; MUFAs: 16:1-CoA, 18:1-CoA (SFA *P* = 0.0168, MUFA *P* = 0.0168). **c**, **d** Cellular proportion of PI- and PC-bound MUFAs (*P* = 0.0000099). **d** Non-targeting siRNA was transfected as control (ctrl). Interconnected lines indicate data from the same independent experiment. **f** Protein expression of BiP. **g** Ratio of LC3BII/LC3BI protein levels. **h** PARP cleavage; cl., cleaved. Western Blots are representative of four (**f**, **h**) and five (**g**) independent experiments. **i** Annexin V and propidium iodide (PI) staining. Proportion of annexin V / PI negative cells (viable cells), annexin V positive / PI negative cells (early apoptotic cells), annexin V positive / PI positive cells (late apoptotic cells), and annexin V negative / PI positive cells (necrotic cells) as percentage of total cells. Data are identical to Fig. 6e. Cytograms are shown in Supplementary Fig. 11b. **j** Fibroblast morphology; scale bar, 100 μm. Phase contrast images are representative of three independent experiments. **k** Cell numbers. **l** PIP₃ levels determined by ELISA. Paired data (**d**) or mean + s.e.m. (**b**, **i**) and single data (**a**, **c**, **e–h**, **k**, **l**) from *n* = 3 (**a**, **d**, **e**, ctrl), *n* = 4 (**b**, **f**, **h**, **i**, **k**, **l**), *n* = 5 (**g**), *n* = 7 (**c**) independent experiments and *n* = 9 (**e**, siRNA) based on three different *Scd1* siRNA in three independent experiments. **P* < 0.05 or *P* values given vs. vehicle control (**a–c**, **f–h**, **k, l**) or control siRNA (**d, e**); two-tailed paired (**a–d**, **f–h**, **k, l**) or two-sided unpaired student *t* test (**e**).

morphology (Fig. 4j) of the same diameter as control cells (Supplementary Fig. 11c), and cell numbers were moderately reduced (Fig. 4k) without cell membrane integrity being impaired (Supplementary Fig. 11d). Notably, PI-derived phosphoinositides (PIPs) were not decreased but rather increased, as exemplary shown for phosphatidylinositol-3,4,5-trisphosphate (PIP₃) (Fig. 4l).

Since tyrosine kinases play a central role in the regulation of the above-mentioned cellular processes, we immunoprecipitated phospho-tyrosine proteins, separated them by SDS gel electrophoresis (Supplementary Fig. 12a), and identified proteins in differentially regulated bands by quantitative proteomics (Supplementary Fig. 12b, Supplementary Data 1). Interestingly, one of the hits, lysosomal group XV phospholipase A₂ (LPLA₂), (i) is activated by negatively charged phospholipids, such as PI, (ii) exhibits specificity for glycerophospholipids with unsaturated fatty acids, including 18:1, both in *sn*−1 and *sn*−2 position, and (iii) transacylates short chain ceramides, which impair proliferation, interfere with ER function and enhance cell death[52]. Together, we show that programmed cell death (i) inhibits SCD1 expression, (ii) reduces MUFA-PI ratios, and (iii) induces p38 MAPK stress signaling, and we demonstrate that SCD1 participates in MUFA-PI biosynthesis and p38 MAPK activation.

By first blocking SCD1 in fibroblasts using CAY10566 and then culturing the cells under diverse cytotoxic conditions, we confirmed a functional link between the early cytotoxic drop in SCD1 (i.e., before programmed cell death is substantially executed) and the above-described phenotypes (Supplementary Figs. 13 and 14, Supplementary Note 4). Moreover, we excluded that p38 MAPK mediates SCD1-dependent changes in the phospholipid composition (Supplementary Fig. 15, Supplementary Note 5) and demonstrated that the early cytotoxic decrease of SCD1 is independent of caspases (Supplementary Fig. 16, Supplementary Note 6).

**MUFA-containing PI predicts stress in *Scd1*-defective mice.** Mice homozygous for the *Scd1*ab-2J allele have a defect *Scd1* gene with an in-frame stop codon in exon 2[53]. To identify SCD1-derived phospholipid species that are inversely regulated to stress signaling in vivo, we focused on organs and tissues that highly express SCD1 and are considered as targets for intervention with SCD1 inhibitors, i.e., liver, skin, hind leg skeletal muscle, and white abdominal fat[24]. Marker proteins of ER stress/UPR and autophagy were substantially elevated in *Scd1*-defective mice. While liver and fat engage both stress-adaptive pathways, only ER stress/UPR is triggered in skeletal muscle and neither is induced in skin (Fig. 5a, Supplementary Fig. 17). Accordingly, levels of phosphorylated p38 MAPK (Thr180/Tyr182) were substantially

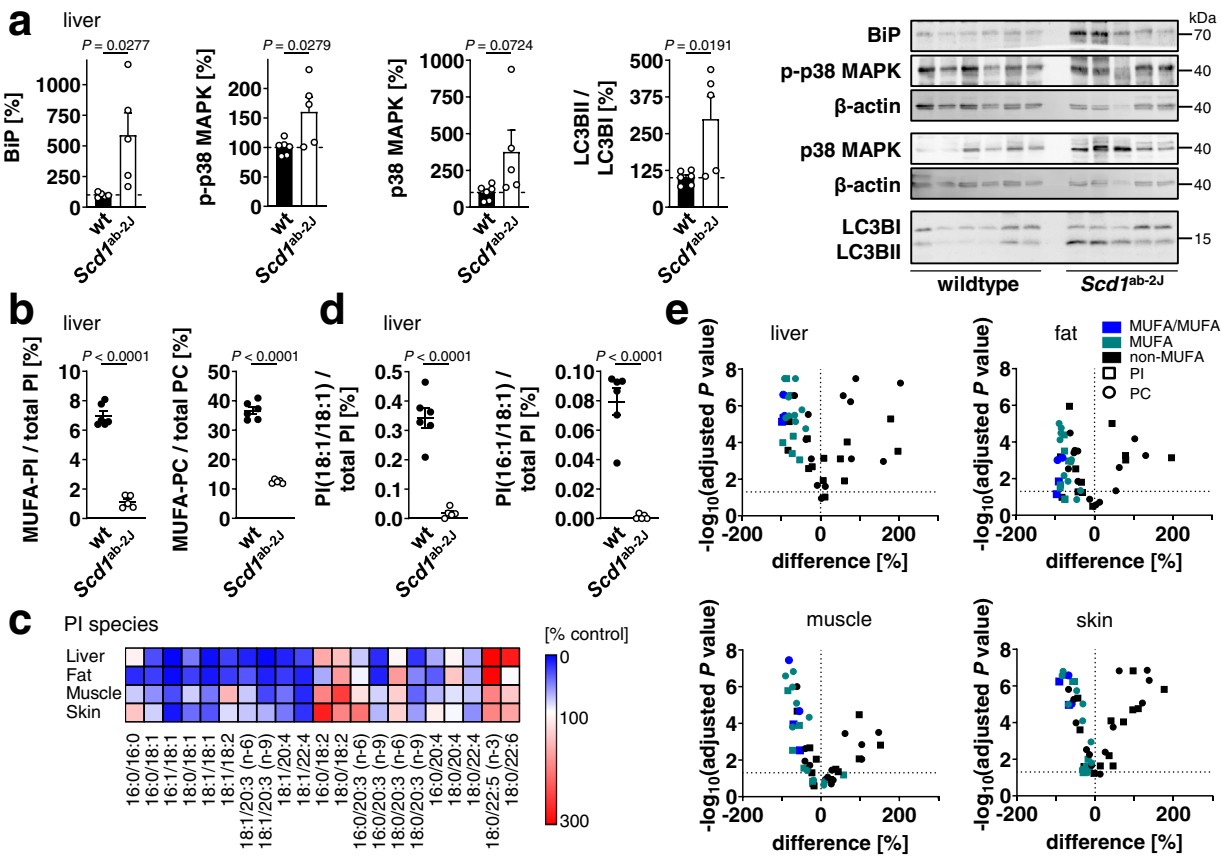

**Fig. 5 MUFA-containing PI species indicate stress in mice with *Scd1* defect.** Liver, white abdominal fat, hind leg skeletal muscle, and skin were obtained from wildtype mice (wt) and mice homozygous for the *Scd1*[ab-2J] allele (*Scd1*[ab-2J]). **a** Protein expression of BiP and p38 MAPK, p38 MAPK phosphorylation, and the ratio of LC3BII/LC3BI protein levels in liver. **b** Hepatic proportion of MUFAs in PI and PC; MUFAs: 16:1, 18:1 (LTR $P = 0.00000008$, $0.00000003$). **c** Heatmap showing the changes in the PI profile of liver, fat, skeletal muscle, and skin from *Scd1*[ab-2J] mice as compared to wt mice. Data are given as percentage of the relative PI abundance in wt tissues. **d** Hepatic proportion of PI(18:1/18:1) and PI(16:1/18:1) (LTR $P = 0.00001$, $0.00003$). **e** Volcano plots highlighting PI and PC species that are regulated in murine tissues from *Scd1*[ab-2J] mice relative to wt mice. Comparisons show the mean difference of percentage changes and the negative log10(adjusted $P$ value). Adjusted $P$ values given vs. vehicle control; two-tailed multiple unpaired student $t$ tests with correction for multiple comparisons using a two-stage linear step-up procedure by Benjamini, Krieger, and Yekutieli (false discovery rate 5%). Phospholipids containing two MUFAs ("MUFA/MUFA") or one MUFA in combination with SFA or PUFA ("MUFA") are indicated by color. Mean (**c**, **e**) or mean + s.e.m. and single data (**a**, **b**, **d**) from $n = 6$ (wt) and $n = 5$ (*Scd1*[ab-2J]; **a** for BiP, wt) mice. $P$ values as indicated; two-tailed unpaired student $t$ test.

higher in liver from *Scd1*[ab-2J] than wildtype mice, which we partly ascribe to an upregulation of total p38 MAPK (Fig. 5a).

Total phospholipid amounts were decreased by *Scd1* inactivation in muscle, not affected in liver, and increased in fat and less in skin (Supplementary Fig. 18a). These tissue-specific differences likely depend on the preferential decrease of neutral lipids[53] and the consequently raising proportion of phospholipids relative to tissue mass. More consistent is the expected drop in MUFA-containing phospholipids in *Scd1*[ab-2J] mice (Fig. 5b, Supplementary Fig. 18b). Among the species that are strongest and most robustly decreased are phospholipids carrying two MUFAs, i.e., combinations of 18:1/18:1 and 16:1/18:1 (Fig. 5c–e and Supplementary Fig. 18c, d). Principal component analysis groups these phospholipids in a tight cluster together with specific MUFA-containing PC and PI species (Supplementary Fig. 18e). The cellular proportion of these clustered phospholipids decreases in tissues from *Scd1*[ab-2J] mice relative to PC and PI species that contain SFAs and PUFAs, such as PI(18:0/22:5n-3), PC(18:0/22:5), PC(18:0/22:6), and PC(18:0/linoleic acid (18:2)), which are substantially upregulated.

PI(18:1/18:1) is preferentially reduced in liver and fat (Fig. 5c), the two tissues that responded most sensitive to the induction of stress-regulated pathways (Fig. 5a and Supplementary Fig. 17). A similar pattern was observed for PI(18:1/18:2), PI(16:0/18:1), PI(18:1/20:3n-6), PI(18:1/20:4) (Fig. 5c) and distinct PC species such as PC(16:1/18:1), PC(18:1/18:1), PC(16:0/16:1), and PC(18:1/20:3) (Supplementary Fig. 18c). Together, *Scd1* inactivation is associated with severe changes in the phospholipid amount and fatty acid composition but only a limited set of species, such as PI(18:1/18:1), predict stress responses across tissues.

**SCD1-derived PI(18:1/18:1) limits stress signaling.** The molecular mechanisms that translate SCD1 activity into biological responses are poorly understood[3]. To investigate whether SCD1-derived phospholipids counteract stress signaling and to identify the signaling lipids involved, we inhibited SCD1 in fibroblasts by CAY10566 and then incorporated defined phospholipid species. We focused on the most abundant PC and PI species that were regulated in apoptotic fibroblasts (Supplementary Fig. 19) and mice defective in *Scd1* (Supplementary Fig. 18e), i.e., the SCD1-derived phospholipids PC(18:1/18:1) and PI(18:1/18:1) with two monounsaturated acyl chains, PC(16:0/18:1) with saturated/monounsaturated acyl chains, and PI(stearic acid (18:0)/20:4) with saturated/polyunsaturated acyl chains. This set was complemented by the saturated phospholipids PC(16:0/16:0) and PI(16:0/16:0) that were used as control.

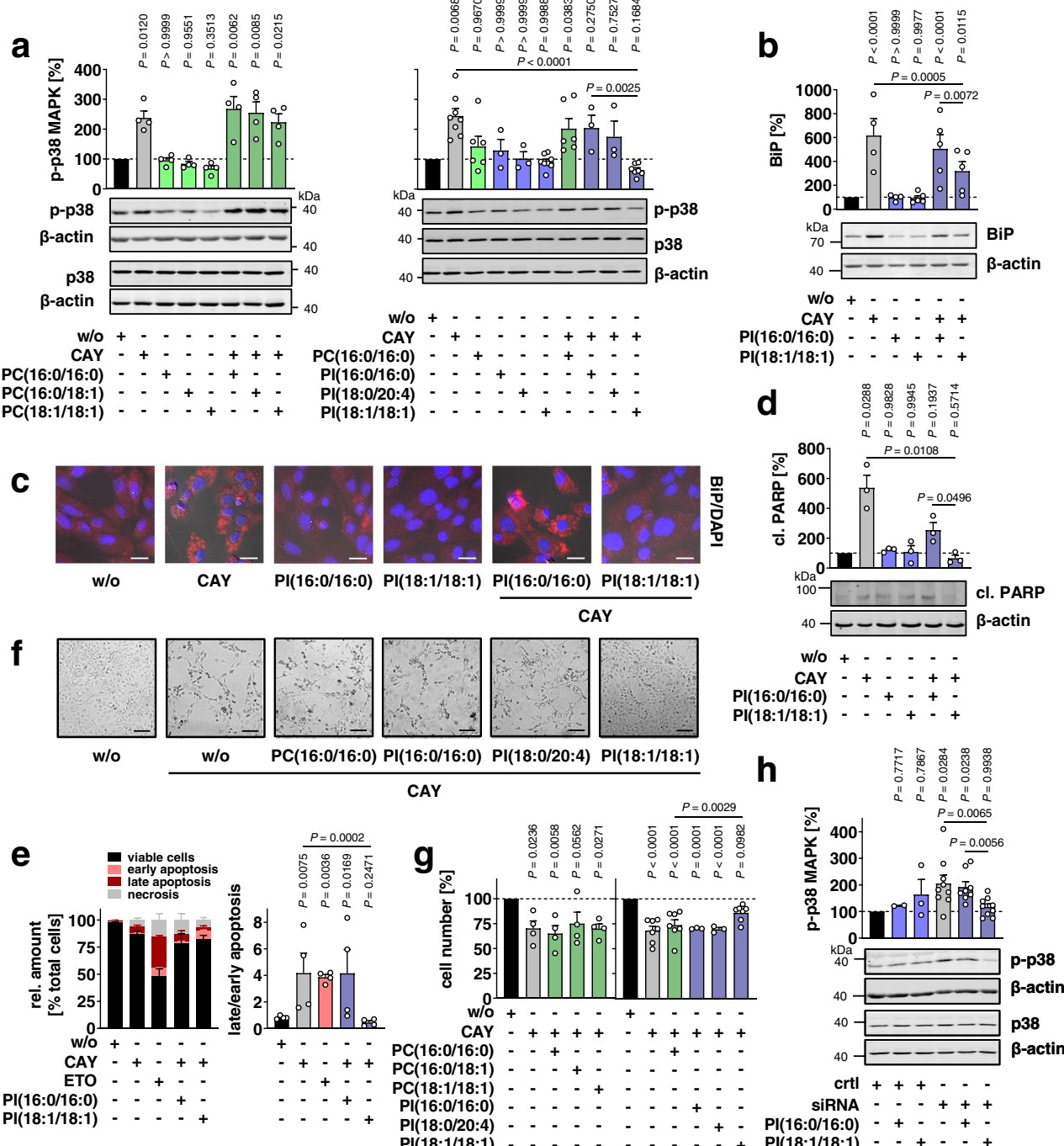

All phospholipids were efficiently taken up by fibroblasts within 48 h when provided as liposomes (Supplementary Fig. 20a), but only PI(18:1/18:1) (and neither the saturated control PI(16:0/16:0) nor PC species) efficiently compensated for the blockage of SCD1: PI(18:1/18:1) increased cellular PI(18:1/18:1) levels above baseline (Supplementary Fig. 20b), prevented p38 MAPK activation (Fig. 6a), reduced ER stress (Fig. 6b, c, Supplementary Fig. 20c), impaired PARP cleavage (Fig. 6d), decreased the ratio of late to early apoptotic cells, indicative of delayed apoptosis (Fig. 6e), and partially restored fibroblast morphology (Fig. 6f and Supplementary Fig. 20d) and cell proliferation in presence of CAY10566 (Fig. 6g). Note that the concentration of supplemented PI(18:1/18:1) (50 μM) is close to the average plasma concentrations for PI(18:1/18:1) and PI(16:1/18:1) in rodents with ad libitum access to food[54].

The compensatory effect of PI(18:1/18:1) on p38 MAPK activation was confirmed by knockdown of SCD1 (Fig. 6h). Alternatively, we added 18:1 to the culture medium and treated cells with CAY10566. Supplementation of 18:1 prevented the decrease in cellular MUFA-PI and PI(18:1/18:1) levels (Supplementary Fig. 21a) and reduced stress signaling (Supplementary Fig. 21b), being comparably effective or even superior to PI(18:1/18:1). The latter particularly applies to UPR induction in CAY10566-treated fibroblasts, where 18:1 (Supplementary Fig. 21b) was more efficient than PI(18:1/18:1) in suppressing BiP expression (Fig. 6b). Supplementation of 16:0 instead potentiated the cytotoxic activity of CAY10566 (Supplementary Fig. 21c). Together, our study reveals prominent stress-suppressive activities for PI(18:1/18:1) but also suggests that PI(18:1/18:1)-independent mechanisms (potentially related to the

**Fig. 6 SCD1-derived PI(18:1/18:1) suppresses stress signaling.** Fibroblasts were treated with vehicle, CAY10566 (CAY, 3 µM), ETO (10 µM), *Scd1* siRNA, and/or defined phospholipid vesicles (50 µM) for 48 h (**a–g**) or 43 h after transfection with siRNA for 5 h (**h**). 16:0 (400 µM) was added 6 h before harvesting (**b–d**). **a** Phosphorylation of p38 MAPK (LTR $P$ = 0.99996, 0.99997, 0.999996, 0.0000004). **b** Protein expression of BiP (LTR $P$ = 0.0000002, 0.9999995, 0.000001). Western Blots are representative of three (**a**, right panel: PI(16:0/16:0) ± CAY, PI(18:0/20:4) ± CAY), four (**a**, left panel), five (**b**), six (**a**, right panel: PC(16:0/16:0) ± CAY) or eight (**a**, right panel: w/o, CAY, PI(18:1/18:1) ± CAY) independent experiments. **c** Subcellular distribution of BiP (red) and DAPI (blue) overlaid with phase contrast images; scale bar, 20 µm. Fluorescence images are representative of three independent experiments. **d** PARP cleavage; cl., cleaved. Western Blots are representative of three independent experiments. **e** Annexin V and propidium iodide (PI) staining. Left panel: viable, early apoptotic, late apoptotic, and necrotic cells were defined as described for Fig. 4i. Data for w/o, CAY and ETO are identical to Fig. 4i. Cytograms are shown in Supplementary Fig. 11b. Right panel: ratio of late to early apoptotic events. **f** Fibroblast morphology; scale bar, 100 µm. Phase contrast images are representative of three independent experiments. **g** Cell numbers (LTR $P$ = 0.000004, 0.000097, 0.00009). **h** *Scd1* was silenced in fibroblasts using three different siRNA sequences, and p38 MAPK phosphorylation was determined after 48 h. Non-targeting siRNA was transfected as control (ctrl). Western blots are representative of three independent experiments. Data for w/o (ctrl and siRNA) are identical to Fig. 4e. Detailed descriptions of datasets shown are given in Supplementary Note 7. $P$ values given vs. vehicle control (**a, b, d, e, g**) or control siRNA (**h**) or as indicated; repeated measures two-way ANOVA (**a**, left panel, **d**), two-way mixed-effects model (REML) (**a**, right panel), repeated measures one-way ANOVA (**e**, right panel, **g**, left panel), or one-way mixed-effects model (REML) (**g**, right panel) of log data or ordinary two-way ANOVA (**b, h**) + Tukey HSD post hoc tests.

---

elevated MUFA/SFA ratio) add to the stress-reducing function of SCD1.

Cellular uptake and distribution of liposomal phospholipids is slow and accompanied by their endocytosis and lysosomal degradation[55]. To exclude that PI(18:1/18:1) markedly degrades and releases MUFAs during this period, we monitored the intracellular concentration of non-esterified fatty acids. While PI(18:1/18:1) strongly accumulated in the cell within 48 h after phospholipid supplementation (Supplementary Fig. 22a), free intracellular 18:1 was not markedly increased (Supplementary Fig. 22b), which further underlines that the phospholipid and not free 18:1 is the critical signaling molecule that mediates the stress-suppressive properties of PI(18:1/18:1).

We used an alternative lipid-delivery strategy based on fusogenic liposomes to instantly incorporate PI(18:1/18:1) into cells. PI(18:1/18:1) was efficiently taken up by fibroblasts as expected (Fig. 7a), but the excess phospholipid was rapidly degraded to baseline within 24 h (Fig. 7b). The transient increase of PI(18:1/18:1) ratios (Fig. 7c) nevertheless blocked long-term p38 MAPK activation (Fig. 7d), diminished the LC3B-II/I ratio indicative of altered autophagy (Fig. 7e), impaired apoptotic PARP cleavage (Fig. 7f), and partially restored cell morphology in CAY10566-treated fibroblasts (Fig. 7g). On the other hand, the single pulse of PI(18:1/18:1) was not sufficient to prevent ER stress (Fig. 7h) and restore cell numbers (Fig. 7i). Thus, PI(18:1/18:1) is a SCD1-derived lipokine, which counteracts major stress-inductive and stress-adaptive responses depending on the kinetics of PI(18:1/18:1) formation and degradation.

**PI(18:1/18:1) biosynthesis and stress signaling.** To gain further insights into the mechanisms that contribute to cytotoxic PI(18:1/18:1) depletion, we quantitatively compared the proteome of VAL- and MC-treated fibroblasts (Supplementary Fig. 23, Supplementary Data 1). Given our focus on overarching cytotoxic mechanisms, we only considered proteins that are concomitantly regulated by VAL and MC in the same direction (Supplementary Data 2). Cytotoxic stress induced by VAL and MC lowered the availability of CTP synthase (Ctps)2 and shifted the balance in the remodeling pathway from PI generation (via lysophospholipid acyltransferase (Lplat)6/Lclat1) towards degradation (via cytosolic phospholipase A$_{2\alpha}$ (Pla2g4a)) (Supplementary Fig. 24a–d). We further investigated whether effects on enzymes in PI biosynthesis and metabolism are mimicked by the SCD1 inhibitor CAY10566 and whether PI(18:1/18:1) compensates for the loss of SCD1 activity. CAY10566 triggered the switch in PI remodeling (Supplementary Fig. 24a, e), and supplementation of PI(18:1/18:1) impaired this catabolic shift more efficiently than the control PI(16:0/16:0) (Supplementary Fig. 24a, b, f). Our findings thus

indicate that PI remodeling is, at least partially, orchestrated by SCD1-derived PI(18:1/18:1). Since Lplat6 accepts 18:1-CoA along with other acyl-CoAs[56] while Pla2g4a prefers PUFA-containing phospholipids[57], it is difficult to estimate whether the counter-regulation of the two enzymes favors PI(18:1/18:1) depletion over other PI species. Note that CAY10566 did not substantially affect the mRNA levels of the terminal enzyme in PI biosynthesis, phosphatidylinositol synthase (PIS) (Supplementary Fig. 24g). The consequences of cytotoxic stress, SCD1 inhibition, and PI(18:1/18:1) on proteins involved in intracellular PI transport and PIP metabolism are illustrated in Supplementary Fig. 24b and summarized in Supplementary Note 8. Together, the altered abundance of proteins in PI biosynthesis and metabolism likely contributes to altered PI levels upon cytotoxic stress, with some of the proteins being regulated by PI(18:1/18:1) or other SCD1-derived metabolites.

To delineate the stress-related effects of PI(18:1/18:1) at proteome levels, we investigated changes in the proteome associated to p38 MAPK signaling, the UPR, autophagy, and programmed cell death (Supplementary Data 3–6). Only proteins were considered, for which CAY10566, VAL, and MC shifted the cellular levels in the same direction, and PI(18:1/18:1) attenuated the effect to a greater extent than the saturated control PI(16:0/16:0). The catalytic subunit of serine/threonine-protein phosphatase 2 A (Ppp2ca) emerged as a major SCD1/PI(18:1/18:1)-regulated protein that depletes under cytotoxic stress (Fig. 8a–c). In combination with regulatory subunits, Ppp2ca dephosphorylates and thus inactivates p38 MAPK, upstream kinases (e.g., MKK3, MKK6)[38–40], and the ER transmembrane serine/threonine-protein kinase/endoribonuclease IRE1α, which senses ER stress and initiates the UPR[41]. Central components of ER-associated degradation (ERAD) were downregulated, among them proteins that participate in the recognition of misfolded proteins (Bcap31), ubiquitination (Skp1), and proteasomal degradation (Psmd12, Psmd13) (Fig. 8a, b). Cytotoxic stress and SCD1 inhibition also substantially increased the abundance of the autophagy receptor sequestosome 1 (Sqstm1, p62) (Fig. 8a–c). p62 is located at the crossroad of UPR, autophagy, and stress-activated kinases and triggers selective autophagy of ubiquitylated cargo, stimulates the activation of initiator caspases, contributes to the assembly of the necroptosis machinery in absence of Map3k7 (which was neither detected under cytotoxic stress nor SCD1 inhibition, Supplementary Data 1), and initiates diverse mitogenic and stress-regulated signaling pathways[58–60].

Supplementary Note 9 describes further changes in the proteome of stressed cells that are counter-regulated by SCD1/PI(18:1/18:1) (Fig. 8a–c and Supplementary Data 3–6).

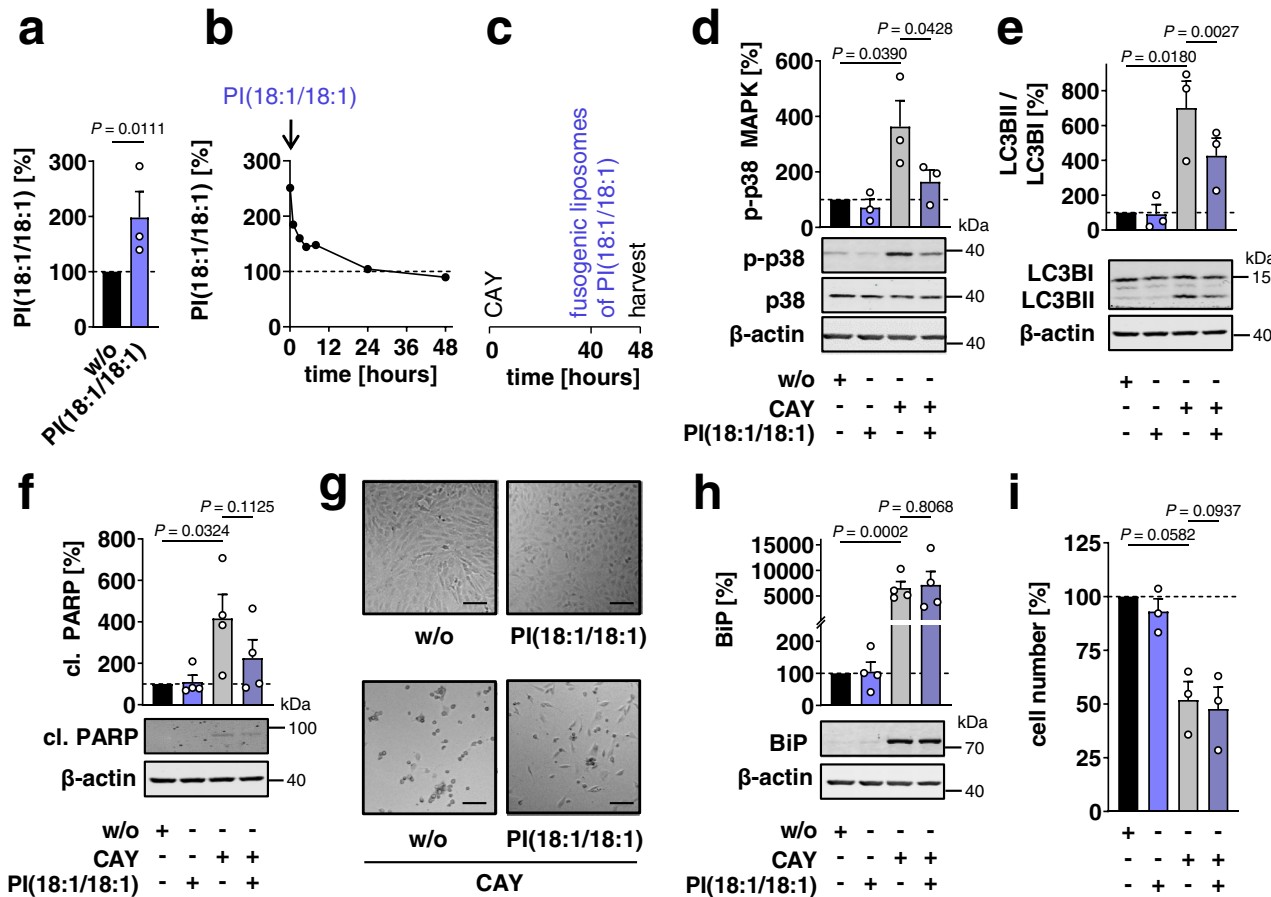

**Fig. 7 PI(18:1/18:1) pulses shape the cellular stress response. a** Cellular content of PI(18:1/18:1) directly after addition of the fusogenic phospholipid mixture with 3 nmol PI(18:1/18:1) to fibroblasts. **b** Time-dependent decrease of excess PI(18:1/18:1). **c–i** Fibroblasts were treated with vehicle or CAY10566 (CAY, 3 μM) for 48 h, and PI(18:1/18:1) (3 nmol) was incorporated into cells using fusogenic liposomes 8 h before harvesting. **c** Time-scale. **d** Phosphorylation of p38 MAPK. **e** Ratio of LC3BII/LC3BI protein levels. **f** PARP cleavage; cl., cleaved. **g** Fibroblast morphology; scale bar, 100 μm. Phase contrast images are representative of three independent experiments. **h** Protein expression of BiP. **i** Cell numbers. Western blots are representative of three (**d**, **e**) or four (**f**, **h**) independent experiments. Blots shown in panel **d** and **f** and **e** and **h** derive from the same membrane, respectively. Mean + s.e.m. and single data from n = 1 (**b**), n = 3 (**a**, **d**, **e**, **i**), n = 4 (**f**, **h**) independent experiments. Two-tailed paired student t test (**a**, **i**) of log data (**d–f**, **h**).

**Stress signaling by death-triggered loss of PI(18:1/18:1)**. The cytotoxic conditions used in our study consistently activated p38 MAPK (Supplementary Fig. 25a) but varied in their effect on the UPR (Supplementary Fig. 25b) and autophagy (Supplementary Fig. 25c). The UPR marker protein BiP was substantially elevated by TPG and VAL and less by MC (Supplementary Fig. 25b), and the central protein in autophagy, LC3B[61], showed strongest PE conjugation for STS, TPG, VAL, and MC (Supplementary Fig. 25c). Our findings confirm that programmed cell death is closely linked to fibroblast stress signaling but also underline the heterogeneity in the response between cytotoxic stressors. To investigate whether the drop of PI(18:1/18:1) during cell death evokes stress signaling, we incorporated the lipid into fibroblasts either using PI(18:1/18:1)-containing liposomes or by supplementing non-esterified 18:1 and then applied cytotoxic conditions. PI(18:1/18:1) decreased (i) the activation of p38 MAPK throughout the cytotoxic settings (Supplementary Fig. 25a), (ii) diminished LC3B lipidation (VAL > TPG, MC > STS) (Supplementary Fig. 25c), and (iii) reduced the UPR in TPG-, VAL- and MC-treated cells by trend (Supplementary Fig. 25b). The suppression of stress signaling by PI(18:1/18:1) is associated with a slightly attenuated apoptotic progression under multiple cytotoxic conditions (Supplementary Fig. 25d) but did not restore viable cell numbers (Supplementary Fig. 25e), as expected from the

failure of PI(18:1/18:1) to decrease the total number of necrotic/ apoptotic cells upon SCD1 inhibition (Fig. 6e). To further explore why supplementation of PI(18:1/18:1) failed to prevent the decrease in cell numbers under cytotoxic conditions, we co-treated fibroblasts with VAL and free 18:1, which is efficiently incorporated into PI and other phospholipids (Supplementary Fig. 21a). While 18:1 neither prevented the overall decrease in viable cell numbers (Supplementary Fig. 27a) nor membrane intactness (Supplementary Fig. 27b), it effectively reduced the proportion of necrotic/apoptotic cells within the fraction of viable (attached) cells, as shown by PI/annexin V staining (Supplementary Fig. 27c, d). Together, the lipokine PI(18:1/18:1) buffers stress responses (i.e., p38 MAPK signaling, UPR) in programmed cell death, tunes survival responses (i.e., autophagy), and, as suggested by 18:1 supplementation studies, seems to be beneficial for the vitality of surviving cells under cytotoxic stress.

**PI(18:1/18:1) in the context of stress-tolerance**. Stress signaling is a central element in (patho)physiological processes, such as tumorigenesis, chemoresistance, aging, and infection. We monitored PI(18:1/18:1) levels in these contexts to explore the phospholipid's potential in adjusting stress tolerance. Our focus was first on malignant cells, whose cancerogenic, metastatic,

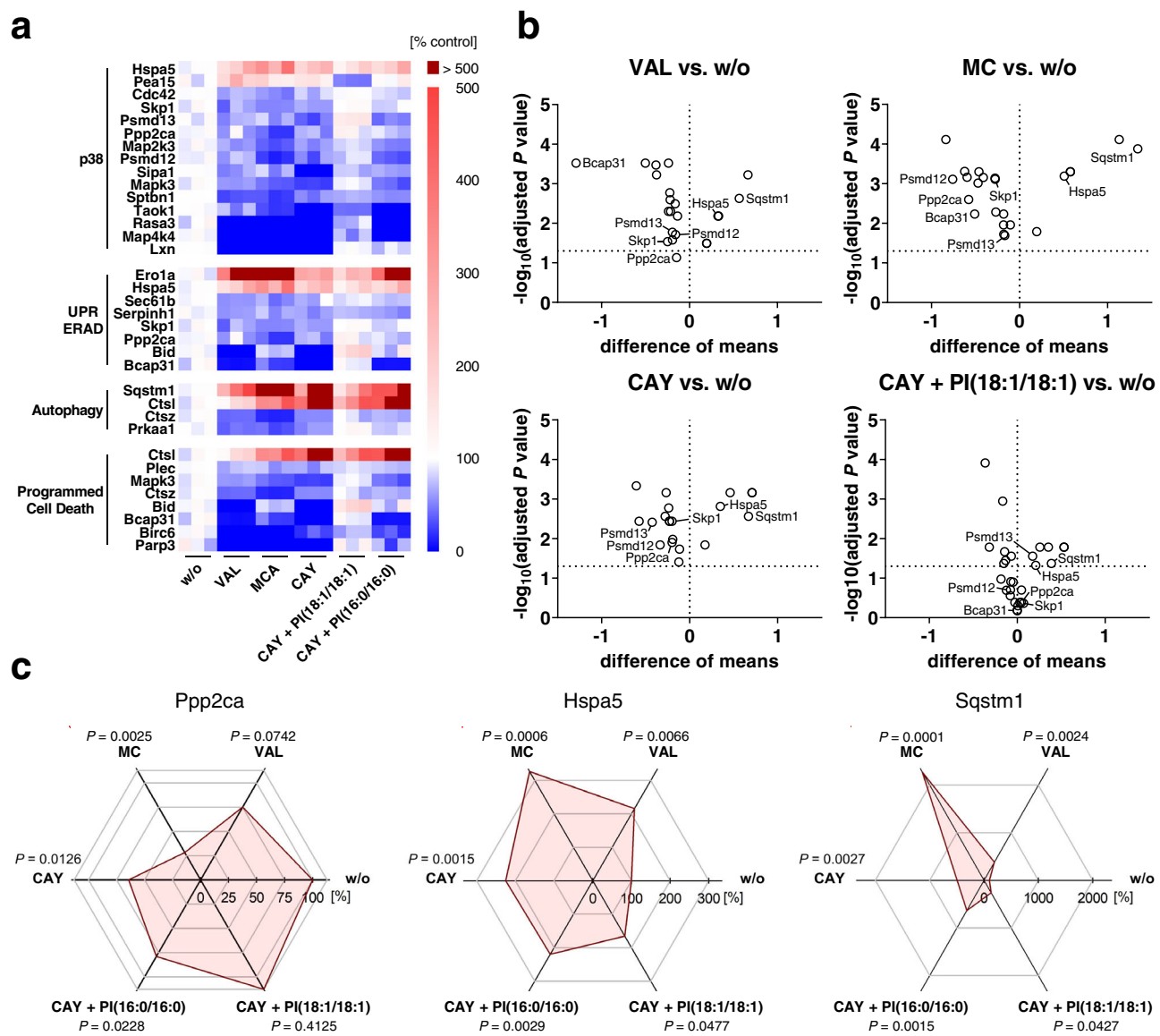

**Fig. 8 Mechanistic insights into stress regulation by PI(18:1/18:1) from quantitative proteomics.** Fibroblasts were treated with vehicle, VAL (10 µM), MC (10 µM), CAY10566 (CAY, 3 µM), CAY (3 µM) plus PI(18:1/18:1) (50 µM), or CAY (3 µM) plus PI(16:0/16:0) (50 µM) for 48 h. **a** Heatmap showing relative changes in protein levels. Focus is placed on proteins that (i) participate in p38 MAPK signaling, the UPR, autophagy, or programmed cell death, (ii) are up- or downregulated by VAL, MC, and CAY10566 in the same direction (≥ 20%), and iii) for which co-treatment with PI(18:1/18:1) diminishes the effect (≥10%) more pronounced than PI(16:0/16:0) (≥ 10% difference). Data of independent experiments ($n = 3$) were calculated as percentage of vehicle control. **b** Volcano plots highlighting proteins that are regulated by VAL, MC, CAY or by PI(18:1/18:1) supplementation in CAY-treated cells compared to vehicle control. Comparisons of the indicated treatment groups show the difference of mean absolute intensities of $\log_{10}$ data and the negative $\log_{10}$(adjusted $P$ value). **c** Radar charts indicating the percentage changes of cellular Ppp2ca, Hspa5, and Sqstm1 in fibroblasts treated with VAL, MC, CAY, CAY + PI(16:0/16:0), or CAY + PI(18:1/18:1) relative to vehicle control. Single data (**a**) or mean (**b**, **c**) from $n = 3$ independent experiments. Adjusted $P$ values given vs. vehicle control; two-tailed multiple unpaired student $t$ tests from log data with correction for multiple comparisons using a two-stage linear step-up procedure by Benjamini, Krieger, and Yekutieli (false discovery rate 5%) (**b**, **c**).

immunosuppressive, and chemoresistant capacity is increased by sustained stress signaling[62]. Notably, high PI(36:2) levels have been reported for tumor cell lines and pancreatic neoplasia in mice and were ascribed to p53 mutations[63]. PI(36:2) comprises multiple isobaric species including PI(18:1/18:1). To substantiate that PI(18:1/18:1) is regulated during tumorigenesis, we investigated B-cell lymphoma from Eµ-Myc-transgenic mice relative to pre-tumoral IgM⁻ B-cells and found a strong accumulation of PI(18:1/18:1) in lymphomas (Fig. 9a). The increase in PI(18:1/18:1) ratios during malignant transformation was associated with enhanced Ulk1 phosphorylation at Ser757 (Fig. 9b). The effect was less pronounced for the IgM⁺ phenotype (Fig. 9a), as

expected from the preferential loss of p53 in IgM⁻ as compared to IgM⁺ lymphomas[64]. Ulk1 participates in the initiation of autophagy and is inhibited by mTORC1 through phosphorylation at Ser757[65]. In agreement with enhanced mTORC1 activity[66], the phosphorylation of further target proteins (4E-BP1, S6-kinase and its substrate rpS6, Fig. 9b) was enhanced. Together, the increase of PI(18:1/18:1) in B-cell lymphoma is associated with elevated mTORC1 activity and inhibition of ULK1, indicative of impaired autophagy.

Sorafenib has been reported to kill cancer cells by inhibiting SCD1-mediated MUFA biosynthesis[67]. To investigate the role of PI(18:1/18:1) in chemoresistance, we generated Huh-7

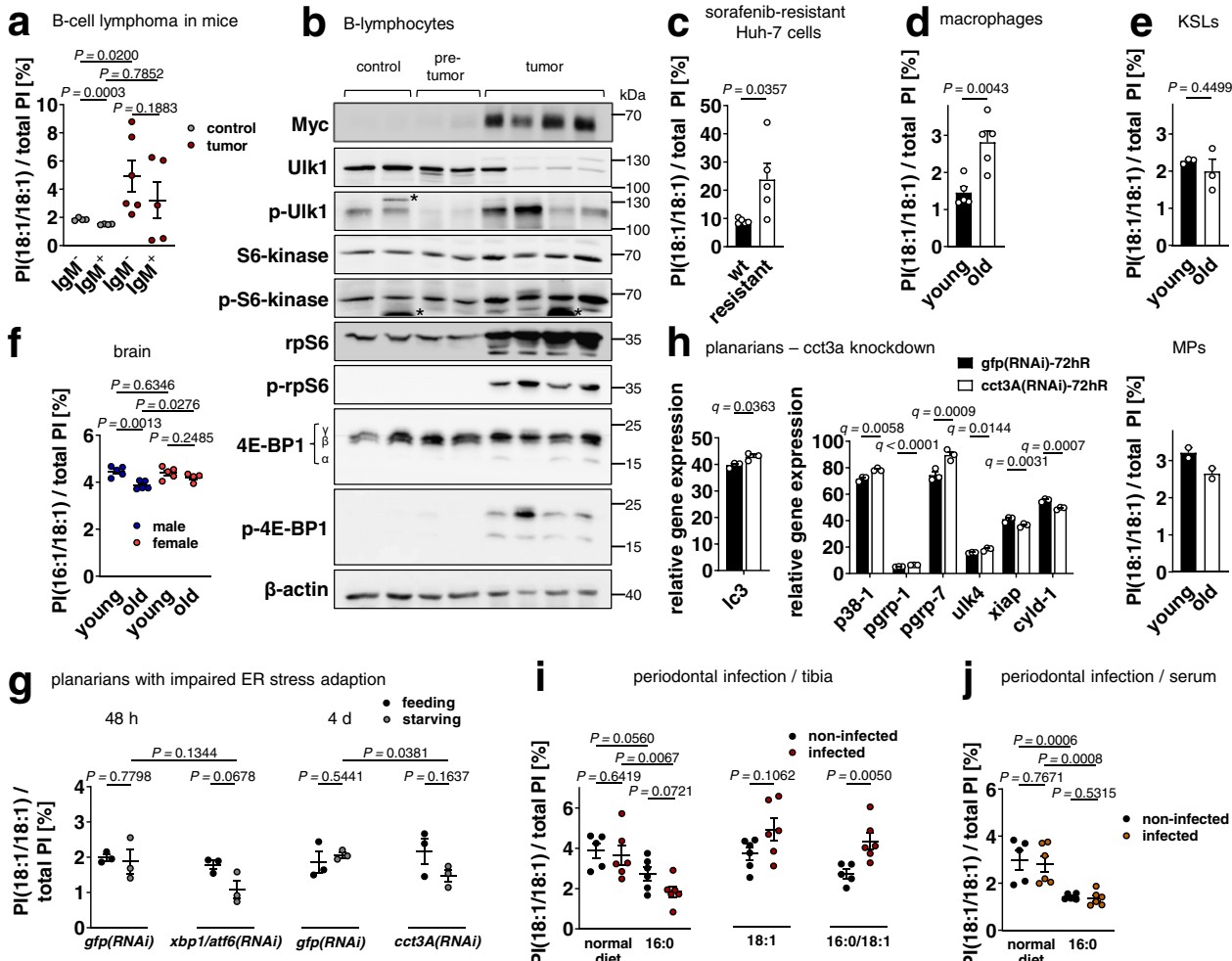

**Fig. 9 PI(18:1/18:1) in systems with varying stress tolerance. a, c–e, g-j** Cellular proportion of PI(18:1/18:1). **a** IgM⁻ and IgM⁺ B-cells and B-cell lymphoma from Eμ-Myc-transgenic mice. **b** Expression, phosphorylation, and substrate turnover of the autophagy-regulatory proteins Ulk1, S6-kinase, and 4E-BP1. Western blots are representative of two independent experiments; * non-specific bands. Each line corresponds to an individual animal. **c** Wild-type ('wt') and sorafenib-resistant ('resistant') Huh-7 hepatocarcinoma cells. **d** Resident peritoneal macrophages from young (6-8 months) and old mice (26 months). **e** Hematopoietic stem cells (KSLs, upper panel) and myeloid progenitor cells (MPs, lower panel) from young and aged mice. **f** Cellular proportion of PI(16:1/18:1) in brain from young (6-8 month) and old male and female mice (26 month). **g, h** Influence of starvation on planarians with impaired ER stress adaption. **g** Lipids were extracted from either 48 h (left panels) or 4 days (right panels) regenerating planarians. *gfp* dsRNA (*gfp(RNAi)*) was injected as control. **h** Relative expression in TPMs of differentially regulated genes related to p38 MAPK activation and stress signaling based on the transcriptome analysis of starved, *gfp* or *cct3A* RNAi-treated planarians at 72 h of regeneration. Significance was determined by q-value (false discovery rate (FDR)) < 0.1 for pairwise comparisons. **i, j** Mice were inoculated with *P. gingivalis* and housed for 16 weeks on normal, 16:0-enriched, or 18:1-enriched diet or first for eight weeks on 16:0-enriched and then eight weeks on 18:1-enriched diet. Mean + s.e.m. and single data from n = 2 (**e**, lower panel), n = 3 (**e**, upper panel, **g**, **h**), n = 4 (**a** for control, **f** for old female), n = 5 (**a** for IgM⁺ tumor, **c**, **d**, **f**, **i**, normal diet for non-infected mice and 16:0/18:1-treated non-infected mice, **j**, normal diet for non-infected mice), n = 6 (**a** for IgM⁻ tumor, **i**, **j**). P values as indicated; two-tailed unpaired student t test (**c**, **d**, **e**, **g**) of log data (**a**, **f**, **i**, **j**).

hepatocarcinoma cells, which are resistant to clinically relevant peak plasma concentrations of sorafenib. p38 MAPK signaling and ER stress are elevated, when these cells are exposed to low sorafenib concentrations[68], and we again found PI(18:1/18:1) levels being markedly increased (Fig. 9c). These observations are likely of clinical relevance because p38 MAPK, whose activation is buffered by PI(18:1/18:1) (Fig. 6a), enhances metabolic stress tolerance and protects mouse hepatocellular carcinoma from sorafenib-induced apoptosis[18].

Aging is associated with an insidious but progressive increase of inflammatory stress, which is closely linked to aberrant immune cell metabolism[69]. We compared peritoneal macrophages from aged and young mice and found strongly increased PI(18:1/18:1) levels in the elderly (Fig. 9d). PI(18:1/18:1) upregulation might be an adaptive strategy of macrophages to

aging-associated low-grade stress that eventually lowers their responsiveness. In line with this hypothesis, aging causes elevated p38 MAPK signaling in macrophages, and selective inhibition of p38 MAPK in aged individuals restores the (macrophage-dependent) capacity to resolve inflammation[70]. On the other hand, PI(18:1/18:1) ratios rather decreased in aged hematopoietic stem cells and myeloid progenitor cells (Fig. 9e), which have elevated stress levels as compared to young cells[71–73] and are more susceptible to programmed cell death by pyroptosis[74].

In brain from aged male mice, PI(18:1/18:1) was hardly detectable but the proportion of the close analog PI(16:1/18:1) substantially decreased (Fig. 9f). The drop in PI(16:1/18:1) correlates with an increase of dysfunctional mitochondria and oxidative stress and a decrease of autophagy in the elderly[75]. Females have lower incidences of distinct neurological diseases,

and their brain has been proposed to be less susceptible to age-related oxidative stress[76]. In support of this hypothesis, the proportion of PI(16:1/18:1) was maintained in aged female brain (Fig. 9f).

We studied the consequences of starvation on the organismal level in planarians. Non-stressed planarians maintained constant PI(18:1/18:1) levels despite of dietary restriction (Fig. 9g). However, starvation markedly reduced the proportion of PI(18:1/18:1) in planarians with diminished ER stress tolerance (Fig. 9g) when either *Smed-xbp1*/*Smed-atf6* or *Smed-cct3A* was silenced[77]. To explore the impact of *Smed-cct3A* on stress signaling, we analyzed the published *cct3A* RNAi transcriptome of starved planarians[77]. The central autophagosomal mRNA *lc3* (PlanMine id: dd_Smed_v6_2838_0_1) was substantially upregulated by *cct3A* RNAi (Fig. 9h). The effect on stress-related genes involved in p38 MAPK activation and apoptosis[78] (Supplementary Table 1) is shown in Fig. 9h and described in Supplementary Note 10. Together, the altered gene expression profile of starved planarians implies that silencing of *cct3A* is associated with an enhanced sensitivity towards p38 MAPK signaling, autophagy and potentially apoptosis, as expected from the depletion of PI(18:1/18:1). It is tempting to speculate that consequent p38 MAPK activation links dietary restriction to cytoprotective gene expression[79].

Infections are another major source for stress signaling in the host, which includes p38 MAPK activation, ER stress, and autophagy[80]. *Porphyromonas gingivalis* (*P. gingivalis*) activates bone-resorbing cells leading to alveolar bone degradation[81]. The effect is enhanced by 16:0-rich diet (Schulze-Späte et al., in preparation) and correlates to plasma SFA levels in obesity[82]. We here show that administration of 16:0 to mice decreases the proportion of PI(18:1/18:1) in tibia, which is more pronounced during infection with *P. gingivalis* (Fig. 9i). Serum PI(18:1/18:1) levels were lower in mice that had received a 16:0 diet in comparison to a normal diet but decreased comparably in healthy and infected mice (Fig. 9j). Supplementation of 18:1 elevated PI(18:1/18:1) ratios preferentially in infected tibia, and the difference between non-infected and infected mice further increased when the first administered 16:0 diet was replaced by a 18:1-enriched diet (Fig. 9i).

Together, tumorigenesis, drug-resistance, starvation, aging, and infection substantially affect the availability of PI(18:1/18:1) in vitro and in vivo with likely impact on stress-tolerance.

## Discussion

Pathways that induce programmed cell death are heterogeneous and differ between cytotoxic triggers[83], which makes the identification of bioactive lipids with general relevance in the death program challenging. By exposing cells to mechanistically diverse cytotoxic conditions and comparing their lipid networks, we here identified the minor glycerophospholipid PI(18:1/18:1) as lipokine that orchestrates stress-inductive and -adaptive signaling (Fig. 10). PI(18:1/18:1) maintains cellular homeostasis, morphology as well as proliferation by (i) suppressing p38 MAPK activation, ER stress, the UPR, and apoptosis, (ii) regulating autophagy, and (iii) activating ERAD. Cellular PI(18:1/18.1) levels are under the control of SCD1, which is repressed by cytotoxic stress through different mechanisms (Supplementary Note 11). The kinetics with which intracellular PI(18:1/18:1) concentrations raise (either leading to sustained availability or a brief pulse) define the cellular responses. Whether early (p38 MAPK-related) stress signaling initiates SCD1 depletion and thus aggravates stress conditions or whether independent mechanisms (triggered by cytotoxic stress) are causative, cannot be fully answered.

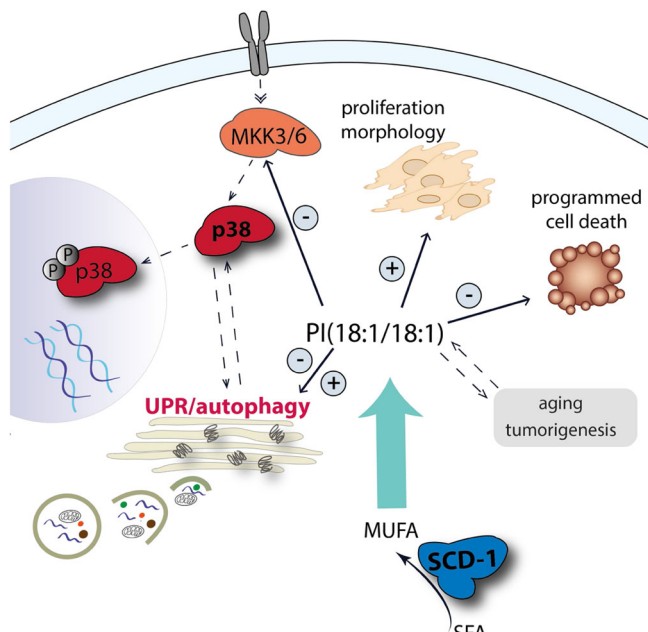

**Fig. 10 Impact of the lipokine PI(18:1/18:1) on stress (-adaptive) signaling.** SCD1-derived PI(18:1/18:1) (i) interferes with the UPR and autophagy, (ii) prevents the activation of p38 MAPK via MKK3/6, (iii) maintains cell morphology and supports cell proliferation, (vi) diminishes the induction of programmed cell death, and (vii) its content is regulated in various stress-related processes such as tumorigenesis, chemoresistance, aging, starvation, and infection.

We confirmed the relevance of our findings for multiple cell lines in vitro and major SCD1-expressing tissues in vivo and provide strong evidence for a prominent role of SCD1-derived PI(18:1/18:1) in reducing stress signaling in programmed cell death. Note that many cytotoxic stressors rapidly activate p38 MAPK with varying mechanisms, kinetics and magnitudes (phase I). The second phase of p38 MAPK activation (and potentially other stress responses) from 6 to 48 h is more consistent and driven by the depletion of SCD1 and PI(18:1/18:1). The directionality and dependency of the stress signaling events are not fully understood, as are the kinetics that underlie other stress responses besides p38 MAPK activation, i.e., the UPR and autophagy. Our conclusions on a common mechanism exclusively refer to the second phase of stress induction, for which we largely obtained good correlations between PI(18:1/18:1) levels and p38 MAPK activation, with only few exceptions that are discussed in Supplementary Note 12. It is tempting to speculate that the stress-protective lipokine PI(18:1/18:1) critically contributes to a variety of (patho)physiological processes besides programmed cell death. In support of this hypothesis, we found that PI(18:1/18:1) is upregulated in cells, tissues, and organisms with increased stress tolerance, i.e., lymphomas, chemoresistant cancer cells, and macrophages from aged mice, and decreased in stressed systems, i.e., starved planarians with restricted stress adaption, aged brain, or periodontal infection under lipotoxic stress.

MUFAs are the major fatty acids in membrane phospholipids and triglycerides[3,24]. They induce mitogenic signaling cascades, enhance proliferation, and reduce apoptosis and/or ferroptosis in various cancer and non-cancer cell lines, whereas SFAs have opposite effects[26,84]. SCD1, which essentially adjusts the cellular ratio of SFAs to MUFAs, improves lipid metabolic profiles and insulin sensitivity[24,85–89], maintains skin homeostasis[90], and

supports tumor growth[91,92], metastasis[93], cancer stemness[94], and chemoresistance[95]. We here show for many cytotoxic conditions that *Scd1* transcription is inhibited by the initiation of programmed cell death and that SCD1 protein levels rapidly decrease, as expected from the N-terminal degradation motif that shortens the half-life of the enzyme[96].

MUFA supplementation and *Scd1* knockout studies propose that the multifaceted biological effects of MUFAs and SCD1 essentially depend on changes in membrane saturation[3]. In fact, SCD1 increases membrane fluidity[3] and rearranges microdomains[97], which has been proposed to influence the function of membrane(-associated) proteins. In support of this hypothesis, phospholipid saturation promotes c-Src clustering within membranes[98] and directly activates ER stress sensors[99,100]. These mechanisms well explain why exogenous SFAs induce stress signaling at lipotoxic concentrations. However, we and others questioned that the physiological regulation of SCD1 evokes sufficient changes in lipid saturation to efficiently modulate cell signaling[3]. An alternative mechanism was described where SCD1 inhibits the accumulation of saturated phosphatidic acid and thus impairs the mineralization of vascular smooth muscle cells[101]. Whether the SCD1-dependent decrease of saturated phosphatidic acid also participates in stress signaling and why 18:1 compensates for SCD1 inhibition is, however, not understood. On the other hand, MUFAs stabilize microdomains that are required for Akt signaling[97], but how SCD1 achieves selectivity over other kinases remained elusive. Interestingly, Akt is anchored to membranes and organized in microdomains through PIPs that are generated by PI phosphorylation[102,103]. SCD1-derived MUFAs further inhibit fatty acid amide hydrolase[104], bind as ligands to receptors such as G-protein-coupled receptor 40[105], and are required as membrane anchors of Wnt proteins[106]. While these studies provide important insights into the pleiotropic functions of SCD1-derived MUFAs, their contribution to stress(-adaptive) signaling is unclear.

Conclusively, the mechanisms how SCD1 and MUFAs prevent stress signaling were enigmatic and demanded for a SCD1-derived signaling lipid that we here describe as PI(18:1/18:1). SCD1 contributes to the biosynthesis of diverse membrane lipids and free fatty acids, and absolute changes in MUFA-containing PC and PE largely exceed the effect on PI(18:1/18:1). However, PI(18:1/18:1) shows the strongest percentage decrease upon SCD1 inhibition. The low abundance combined with high susceptibility qualifies PI(18:1/18:1) for an effective signaling molecule, whose levels can be rapidly and substantially altered. Whether other MUFA-PIs like PI(16:1/18:1) share the lipokine activity of PI(18:1/18:1) is still elusive, as discussed in Supplementary Note 13.

Mechanistically, we show that the cytotoxic depletion of SCD1 decreases the cellular availability of the catalytic subunit of the serine/threonine-protein phosphatase Ppp2ca partially via PI(18:1/18:1). Among the pleiotropic substrates of Ppp2ca are central players in p38 MAPK activation, UPR induction, and autophagy initiation[107]. The ability of PI(18:1/18:1) to maintain Ppp2ca expression under cytotoxic conditions might thus provide a rational for the phospholipid's stress-protective properties. Note that not only the catalytic subunit but also the protein phosphatase 2 scaffold subunit Aα (Ppp2r1a) is substantially decreased by cytotoxic stress and SCD1 inhibition (Supplementary Data 1). Further clarification is required regarding the order of signaling events, the cytotoxic mechanisms of Ppp2ca repression, and the contribution of Ppp2ca to transmit the signal from PI(18:1/18:1) to stress (-adaptive) pathways (Supplementary Note 7). Whether Ppp2r1a or other regulatory B subunits define Ppp2ca activity and substrate specificity[107] also remains enigmatic.

While our data strongly support that SCD1-derived PI(18:1/18:1) regulates autophagy, further studies are required to define the directionality. On the one hand, we show that CAY10566 enhances the levels of LC3B-II, which is critical for autophagosome formation[8]. On the other hand, proteomic profiling shows increased levels of the autophagy receptor p62. Both LC3 and p62 exhibit enhanced expression and degradation upon autophagy induction, and, thus, our observations can result from decreased or increased autophagic flux[108]. This balance might also provide a rational for seemingly contradictory findings on MUFA supplementation and SCD1 inhibition, which have been proposed to either increase or decrease the autophagic flux[109]. Interestingly, Ppp2ca forms a complex with B55α and dephosphorylates ULK1 (S637)[42], which might explain the autophagy-regulatory activity of PI(18:1/18:1) under cytotoxic stress.

PIPs regulate cell physiology as key factors in signal transduction, actin dynamics, and membrane trafficking, with the biological function defined by the phosphorylated headgroups[110]. Considering the broad spectrum of PIP activities, we wondered whether PI(18:1/18:1) is an inactive precursor that is activated by inositol phosphorylation. Such a transformation of PI(18:1/18:1) into PIPs might be accelerated by 18:1, which was described to induce PI3K activity in breast cancer cells[26]. In analogy, we here found that SCD1-derived PI(18:1/18.1) is important to maintain the basal availability of phosphatases and phospholipases involved in PIP degradation. Further studies are needed to explore the consequences on PIP subclasses.

In conclusion, we identified PI(18:1/18:1) as SCD1-derived lipokine, which maintains cell homeostasis, morphology and survival by suppressing p38 MAPK stress signaling and limiting stress responses, including the UPR and autophagy. This mechanism engages major stress-regulatory proteins, including PPP2CA, MLK3/MKK3/6, and p62, and seems to be of relevance for different cell types and tissues independent of the cytotoxic trigger. PI(18:1/18:1) is upregulated in stress-tolerant systems, e.g., aged immune cells, chemoresistant cancer cells, and tumors, and downregulated in stress-sensitive systems, e.g., aged hematopoietic stem cells and brain, starved planarians lacking adaptive stress mechanisms, and infected bone under lipotoxic stress. Moreover, we provide strong evidence that the early cytotoxic depletion of PI(18:1/18:1) enhances stress(-adaptive) responses in programmed cell death. Our data suggest PI(18:1/18:1) turnover as valuable target in stress-related diseases like the metabolic syndrome or cancer and point towards putative side effects when interfering with SCD1 activity.

## Methods

**Materials**. CHX, ETO, VAL, I3M, and CAY10566 were purchased from Cayman Chemical (Ann Arbor, MI). STS was bought from Calbiochem (Darmstadt, Germany), TPG was from Enzo Life Sciences (Farmingdale, NY), and Q-VD-OPh was from Sigma-Aldrich (St. Louis, MO). Soraphen A, MC and Skepinone-L were kindly provided by Dr. Rolf Müller (Saarland University, Germany), Dr. Johann Jauch (Saarland University, Germany), and Dr. Stefan Laufer (University Tübingen, Germany), respectively. Apoptotic inducers and enzyme inhibitors (purity ≥ 95%) were dissolved in DMSO and stored in aliquots under argon and protected from light at −80 °C.

TNFα was obtained from Peprotech (Hamburg, Germany) and diluted in PBS pH 7.4. Phospholipids and fatty acids were purchased from Avanti Polar Lipids (Alabaster, AL), Cayman Chemical or Sigma-Aldrich and dissolved in chloroform, and aliquots were stored at −80 °C under argon.

Rabbit anti-4E-BP1 (1:1000; # 9452), rabbit anti-BiP (C50B12; 1:500 − 1:1000; # 3177), mouse anti-cleaved PARP (Asp214; 7C9; 1:500 − 1:1000; # 9548), rabbit anti-c-Myc (D84C12; 1:1000; # 5605), rabbit anti-GAPDH (14C10; 1:1000; # 2118), rabbit anti-MKK3 (D4C3; Dilution: 1:1000; # 8535), rabbit anti-MKK6 (D31D1; 1:1000; # 8550), rabbit anti-p38 MAPK (D13E1; 1:500 − 1:1000; # 8690), rabbit anti-p70 S6-kinase (E8K6T; 1:1000; # 9202), and rabbit anti-phospho-4E-BP1 (Thr37/46; 1:1000; # 9459), rabbit anti-phospho-MKK3 (Ser189)/MKK6 (Ser207) (D8E9; 1:1000; # 12280), rabbit anti-phospho-p38 MAPK (Thr180/Tyr182; 1:500 − 1:1000; # 9211), rabbit anti-phospho-p70 S6-kinase (Thr389; 108D2; 1:1000; # 9234), rabbit anti-phospho-S6 Ribosomal Protein (Ser235/236; 2F9; 1:1000; # 4856), rabbit anti-phospho-SAPK/JNK (Thr183/Tyr185; 81E11; 1:1000; # 4668), rabbit anti-phospho-SEK1/MKK4 (Ser257; C36C11; 1:1000; # 4514), rabbit anti-phospho-Ulk1 (Ser757; 1:1000; # 6888), rabbit

anti-S6 Ribosomal Protein (5G10; 1:1000; # 2217), rabbit anti-SAPK/JNK (1:1000; # 9252), rabbit anti-SCD1 (M38; 1:1000; # 2438), rabbit anti-β-actin (13E5; 1:1000; # 4970), and mouse anti-β-actin (8H10D10; 1:1000; # 3700) were obtained from Cell Signaling Technology (Danvers, MA). Rabbit anti-LC3B (1:1000; # ab48394), mouse anti-p38 MAPK (M138; 1:1000; # ab31828), rabbit anti-phospho-MLK3 (Thr277, Ser281; 1:500 − 1:1000; # ab191530), and rabbit anti-Ulk1 (EPR4885(2); 1:1000; # ab128859) were from Abcam (Cambridge, UK) and rabbit anti-β-actin (1:10,000; # A2066) was purchased from Sigma-Aldrich. Secondary antibodies were obtained from LI-COR Biosciences (Lincoln, NE) or Thermo Fisher Scientific (Waltham, MA).

**Cell culture**. Cells were cultivated at 37 °C and 5% CO$_2$. Cell lines were from the DSMZ-German Collection of Microorganisms and Cell Cultures (DSMZ, Braunschweig, Germany), the American Type Culture Collection (ATCC, Manassas, VA) or the Japanese Collection of Research Bioresources Cell Bank (JCRB Cell Bank, Ibaraki, Japan).

Adherent cell lines: Cells were harvested with trypsin/EDTA (GE Healthcare, Munich, Germany or Sigma-Aldrich) and reseeded every three to four days. Mouse NIH-3T3 fibroblasts (# ACC 59, DSMZ; $5 \times 10^5/25$ cm$^2$) were cultivated in DMEM high glucose medium (4.5 g/l; Merck, Darmstadt, Germany or Thermo Fisher Scientific) containing heat-inactivated fetal calf serum (FCS, 10%; Sigma-Aldrich). Human HeLa cervical carcinoma cells (# ACC 57, DSMZ; $4 \times 10^5/25$ cm$^2$) were grown in DMEM supplemented with FCS (10%), penicillin/streptomycin (100 U/ml and 100 µg/ml; GE Healthcare), and L-glutamine (2 mM; Sigma-Aldrich). RPMI 1640 medium (Sigma-Aldrich) with the above-mentioned supplements was used for cultivation of human HT-29 colon adenocarcinoma cells (# HTB-38, ATCC), human HEK-293 embryonic kidney cells (# CRL-1573, ATCC), and human HepG2 hepatoma cells (# ACC 180, DSMZ), which were each seeded at 4 to $5 \times 10^5/25$ cm$^2$. Human MCF-7 breast adenocarcinoma cells (# HTB-22, ATCC; $4 \times 10^5/25$ cm$^2$) were cultivated in DMEM plus FCS (10%), penicillin/streptomycin (100 U/ml and 100 µg/ml), and 0.1% insulin (Sigma-Aldrich). Human HUH-7 hepatocarcinoma cells (# JCRB0403, JCRB Cell Bank; 4 to $5 \times 10^5/25$ cm$^2$) were maintained in DMEM plus FCS (10%) and penicillin/streptomycin (100 U/ml and 100 µg/ml).

Suspension cell lines: Human MM6 acute monocytic leukemia cells (# ACC 124, DSMZ; $1.2 \times 10^6$ cells/4 ml) were grown in RPMI 1640 containing FCS (10%), penicillin/streptomycin (100 U/ml and 100 µg/ml), L-glutamine (2 mM), oxalic acid (1 mM; Sigma-Aldrich), sodium pyruvate (1 mM; GE Healthcare), and non-essential amino acids (1×; Sigma-Aldrich).

Chemoresistant cell line: To prepare and characterize chemoresistant HUH-7 hepatocarcinoma cells, resistant HUH-7 cells were obtained by exposing HUH-7 cells to increasing doses of sorafenib up to 10 µM in the culture medium[68]. The medium of sorafenib-resistant HUH-7 cells was supplemented with sorafenib (10 µM; BAY 43 9006, Enzo Life Sciences GmbH, Lörrach, Germany) to maintain resistance.

Human primary cells: HUVECs from human umbilical cord veins were kindly provided by Dr. Alexander Mosig (University Hospital Jena, Germany)[111] and seeded at $1.5 \times 10^5$ cells/cm$^2$ in Endothelial Cell Medium (ECM) (Promocell, Heidelberg, Germany) for cultivation up to passage 4. Human monocytes ($1 \times 10^7$ cells/4 ml) were isolated from leukocyte concentrates and immediately used for experiments. Leukocyte concentrates were provided by the Institute for Transfusion Medicine of the University Hospital Jena (Germany) from healthy adult volunteers on informed consent[112], and experiments were approved by the ethical commission of the Friedrich-Schiller-University Jena.

Cell treatment: Cells were seeded under the conditions specified above unless indicated otherwise. After cultivation for 24 h, cells were treated with vehicle, TNFα (10 ng/ml), STS (0.3 µM), CHX (20 µg/ml), ETO (10 µM), TPG (2 µM), VAL (10 µM), MC (10 µM), or I3M (10 µM) and/or inhibitors at 37 °C and 5% CO$_2$. For serum depletion of NIH-3T3 fibroblasts, the culture medium was replaced against serum-free DMEM 24 h after seeding. Detached cells were recovered from the cell culture medium to analyze the number of late apoptotic, necrotic, and dead cells, except stated otherwise. In all other experiments, detached cells were discarded and fibroblasts were washed with PBS pH 7.4 to enrich viable and early apoptotic cells. Human monocytes were cultivated under cytotoxic conditions in RPMI 1640 medium containing FCS (10%), penicillin/streptomycin (100 U/ml and 100 µg/ml), and L-glutamine (2 mM). Cell lines used in this study were tested for mycoplasma, and MCF-7 and HEK-293 cells were authenticated. The authentication was performed by Multiplexion (Friedrichshafen, Germany; December, 2020) using Single Nucleotide Polymorphism (SNP) profiling (Multiplex Cell Line Authentication, https://www.multiplexion.de/en/cell-line-testing-service/multiplex-human-cell-line-authentication). DNA for SNP profiling was isolated from cell pellets using an innuPREP DNA Mini Kit (Analytik Jena) according to the manufacturer's instructions. Other cell lines were not authenticated. Cell morphology of all cell lines was regularly inspected. HEK-293 is described as misidentified cell line, which we studied because of the high basal p38 MAPK phosphorylation.

**Extraction of lipids**. Phospholipids and fatty acids were extracted from cell pellets, plasma, planarians, and tissue homogenates by successive addition of PBS pH 7.4, methanol, chloroform, and saline to a final ratio of 14:34:35:17[113,114]. Evaporation of the organic layer yielded a lipid film that was dissolved in methanol and subjected to UPLC-MS/MS. Internal standards: 1,2-dimyristoyl-sn-glycero-3-phosphatidylcholine, 1,2-dimyristoyl-sn-glycero-3-phosphatidylethanolamine, 1,2-di-heptadecanoyl-sn-glycero-3-phosphatidylglycerol, 1,2-diheptadecanoyl-sn-glycero-3-phosphoserine, 1,2-dioctanoyl-sn-glycero-3-phospho-(1'-myo-inositol), and (15,15,16,16,17,17,18,18,18-d9)oleic acid.

For the extraction of acyl-CoAs, NIH-3T3 cell pellets were taken up in aqueous methanol (70%), and proteins were precipitated for 1 h at −20 °C[115]. Samples were vigorously mixed, adjusted to methanol/water (50/50), and incubated for 1 h at −20 °C. After centrifugation (20,000×g, 5 min, 4 °C), the supernatant was evaporated to dryness and extracted with aqueous methanol (50%). Internal standard: [$^{13}C_3$]-malonyl-CoA (Sigma-Aldrich).

**Lipid analysis by reversed phase UPLC and ESI tandem MS**. Chromatographic separation of phospholipids and fatty acids was carried out on an Acquity UPLC BEH C8 column (1.7 µm, 2.1 × 100 mm, Waters, Milford, MA) using either an Acquity$^{TM}$ (Waters)[116] or ExionLC$^{TM}$ AD UHPLC (Sciex, Framingham, MA). The ExionLC$^{TM}$ AD UHPLC was operated at a flow rate of 0.75 ml/min using mobile phase A (acetonitrile/water, 95/5, 2 mM ammonium acetate) and mobile phase B (water/acetonitrile, 90/10, 2 mM ammonium acetate). The gradient was ramped from 75 to 85% mobile phase A within 5 min and to 100% mobile phase A within 2 min followed by isocratic elution for another 2 min. The column temperature was kept at 45 °C. Eluted lipids were detected by multiple reaction monitoring using either a QTRAP 5500 or QTRAP 6500$^+$ Mass Spectrometer (Sciex) equipped with an electrospray ionization source[29,117]. Fatty acid anions of glycerophospholipids were detected in the negative ion mode by multiple reaction monitoring (MRM)[21]. For the QTRAP 6500$^+$ Mass Spectrometer, the curtain gas was set to 40 psi, the collision gas to medium, the ion spray voltage to 4500 V in negative mode, the heated capillary temperature to 350 (PC) – 500 °C (PI), the sheath gas pressure to 55 psi, and the auxiliary gas to 75 psi. The more intensive transition was used for quantitation. Sphingomyelins were analyzed in the positive ion mode based on the detection of the choline headgroup (m/z = 184) by MRM (collision energy: 33 eV). Free fatty acids were quantified in the negative ion mode by single ion monitoring.

For the analysis of triglycerides and cholesteryl ester, acetonitrile/water (95/5, 10 mM ammonium acetate, mobile phase A, 100%) and isopropanol (mobile phase B) were ramped from 100% A to A/B = 70/30 over 6 min followed by isocratic elution for 4 min[29]. Multiple reaction monitoring addressed the fragmentation of [M + NH$_4$]$^+$ adduct ions to [M - fatty acid anion]$^+$ ions, without differentiating between fatty acid positional isomers. The collision energy was set to 35 eV (triglycerides) or 22 eV (cholesteryl ester) and the declustering potential to 55 (CE) − 120 V (TAG).

Acyl-CoA ester of SFAs and MUFAs were separated on an Acquity$^{TM}$ UPLC BEH C18 column (1.7 µM, 2.1×50 mm) and detected by MRM based on the neutral loss of 2'-phospho-ADP ([M + H-507]$^+$) in the positive ion mode[115]. The collision energy was set to 45 eV and the declustering potential to 60 V.

Absolute lipid quantities refer to the internal standard of the subclass and are normalized to cell numbers, protein content, or tissue weight. Relative proportions of lipids are given as percentage of the sum of all species detected in the respective subclass (=100%). To calculate the cellular proportion of phospholipid-bound MUFAs, we summarized the relative intensities of phospholipid species that contain at least one fatty acid with a single double bond. Mass spectra were obtained and processed using Analyst 1.6 or 1.7 (Sciex)[117].

**SDS-PAGE and western blotting**. Cells were resuspended in lysis buffer [20 mM Tris-HCl (pH 7.4), 150 mM NaCl, 2 mM EDTA, 1% (v/v) Triton X-100, 5 mM sodium fluoride, 10 µg/ml leupeptin, 60 µg/ml soybean trypsin inhibitor, 1 mM sodium vanadate, 2.5 mM sodium pyrophosphate, and 1 mM phenylmethanesulphonyl fluoride] and sonified (2 × 5 s, on ice). Murine tissues (100 mg/mL) were homogenized using an Omni tissue homogenizer TH (Kennesaw, GA) or POLY-TRON PT 1200 E Homogenizer (Kinematica, Eschbach, Germany) in lysis buffer. Samples were centrifuged once (12,000×g, 5 min, 4 °C) for cellular samples or twice for murine tissues (15,000×g, 10 min, 4 °C). The protein concentrations of the supernatants were analyzed by DC protein assay kit (Bio-Rad Laboratories GmbH, Munich, Germany). Supernatants were adjusted to same protein concentrations, mixed with 1 × SDS/PAGE sample loading buffer [125 mM Tris-HCl pH 6.5, 25% (m/v) sucrose, 5% SDS (m/v), 0.25% (m/v) bromophenol blue, and 5% (v/v) β-mercaptoethanol], and heated for 5 min at 95 °C. Aliquots (10-20 µg) were separated by 10-15% SDS-PAGE and transferred to a Hybond ECL nitrocellulose membrane (GE Healthcare, Munich, Germany). Membranes were blocked with 5% (m/v) BSA or skim milk for 1 h at room temperature and incubated with primary antibodies overnight at 4 °C.

As secondary antibodies, IRDye 800CW-labeled anti-rabbit or anti-mouse IgG (1:10,000, each; anti-mouse: # 926-32210; anti-rabbit: # 926-32211; LI-COR Biosciences), IRDye 680LT-labeled anti-rabbit or anti-mouse IgG (1:80,000, each; anti-mouse: # 926-68020; anti-rabbit: # 926-68021; LI-COR Biosciences), DyLight® 800 anti-rabbit IgG (1:10000; # SA510036; Thermo Fisher Scientific), and/or DyLight® 680 anti-mouse or anti-rabbit IgG (1:10000, each; anti-mouse: # 35519; anti-rabbit: # 35569; Thermo Fisher Scientific) were used. Fluorescence signals were detected by an Odyssey infrared imager (LI-COR Biosciences) or Fusion FX7 Edge Imaging System (spectra light capsules: C680, C780; emission filters: F-750, F-850; VILBER Lourmat, Collegien, France). Data from densitometric analysis were linearly adjusted and background-corrected, and protein levels were normalized to β-actin (Odyssey system). When using the Fusion FX7 system, densitometric

analysis was performed with Evolution-Capt Edge Software Version 18.06 (VILBER Lourmat) using rolling ball background-subtraction, and protein levels were normalized to β-actin or GAPDH.

Membranes from independent data sets were normalized to total band intensities to compensate varying scanning settings.

Uncropped blots are shown in the source data (main figures) and Supplementary Figs. 28–37 (supplementary figures).

**Analysis of cellular PIP$_3$ levels**. NIH-3T3 fibroblasts ($1.5 \times 10^6/75$ cm$^2$) were cultured for 24 h at 37 °C and 5% CO$_2$. After treatment with vehicle or CAY10566 for 48 h, PIP$_3$ was extracted and quantified using a PIP3 Mass ELISA Kit (K-2500s; Echelon Biosciences Inc., Salt Lake City, UT). In brief, the culture medium was discarded, and cells were immediately treated with ice cold 0.5 M trichloric acid (TCA), centrifuged and washed twice with 5% TCA/1 mM EDTA. After neutral lipids were removed from the pellet by methanol/chloroform (2:1), acidic lipids (including PIP$_3$) were extracted by methanol/chloroform/12 N HCl (80:40:1) and subsequently from the supernatant by chloroform/0.1 N HCl (36:64). PIP$_3$ concentrations were determined according to the manufacturer's instructions and normalized to the protein content. Protein levels were determined using a DC protein assay kit (Bio-Rad Laboratories GmbH) in TCA-acidified cellular samples (100 µl) after neutralization with NaOH (1 M, 50 µl) and addition of aqueous Triton X-100 (10%, 15 µl). Absorbance was measured using a SpectraMax iD3 Microplate Reader which was operated by SoftMax Pro 7.1 (Molecular Devices, San José, CA).

**Mice**. Mice were housed in a controlled environment (temperature: 22 ± 2 °C; humidity: 40–70%) and provided with standard rodent chow and water. Animals were subjected to 12 h light/12 h dark schedule.

**Aged murine resident peritoneal macrophages and brain**. To study PI(18:1/18:1) ratios in resident peritoneal macrophages and brain, we kindly received male and female C57BL/6JRj mice (26 month) from the animal facility of the Leibniz Institute (Jena, Germany). Young male and female C57BL/6JRj mice (6-8 month) were bought from Janvier Labs (Le Genest-Saint-Isle, France). Mice were killed by CO$_2$ inhalation. Resident peritoneal macrophages were obtained by lavage of the peritoneal cavity with 7 ml cold DMEM containing heparin (5 U ml$^{-1}$).

**Organs and tissues of mice with defective Scd1 mutation**. Male C57BL/6 mice (6 weeks) were obtained from the animal facility of the Leibniz Institute (Jena, Germany), killed in a saturated CO$_2$ atmosphere, and hind leg skeletal muscle, white abdominal fat, liver, and skin were collected. Respective tissues from male C57BL/6 J mice homozygous for the Scd1$^{ab-2J}$ allele (6 weeks) were purchased from Jackson Laboratory (Bar Harbor, ME).

Mouse organs, tissues and resident peritoneal macrophages were homogenized in lysis buffer for Western blot analysis or in PBS pH 7.4 for lipid extraction and UPLC-MS/MS analysis.

**Isolation of hematopoietic stem and progenitor cells**. Bone marrow cells, which were freshly isolated from tibia and femur of male and female young (3 to 6 month) and old (18 to 24 month) C57BL/6 mice, were enriched by magnetic activated cell separation (Miltenyi Biotech, Bergisch Gladbach, Germany), immunolabeled with Sca-1 and lineage antibodies and sorted by FACS[118]. In brief, crushed bone samples were incubated with allophycocyanin (APC)-conjugated anti-mouse c-Kit (clone 2B8; 1:100; # 17-1171; Thermo Fisher Scientific) for 30 min at 4 °C and combined after washing with anti-APC microbeads (Miltenyi Biotech). cKit$^+$ cells were magnetically enriched using a LS column (Miltenyi Biotech) and incubated with lineage cell detection cocktail-biotin (# 130-092-613, Miltenyi Biotech) for 30 min at 4 °C and with APC anti-mouse c-Kit (clone 2B8; 1:100; # 17-1171; Thermo Fisher Scientific), PE/Cy7 anti-mouse Sca-1 (clone E13-161.7; 1:100; # 122513; BioLegend, San Diego, CA), and APC/Cy7-conjugated streptavidin (1:100; # 405208; BioLegend) overnight at 4 °C. After staining with DAPI, Lin$^-$ cKit$^+$ ScaI$^+$ (KSL) cells and Lin$^-$ cKit$^+$ ScaI$^-$ myeloid progenitor (MP) cells were sorted using a FACSAria II instrument (BD Biosciences, Franklin Lakes, NJ). FlowJo software was used for data analysis.

**Generation and collection of B-cell lymphomas**. B-cells from 8-week-old male wild-type or Tg(IghMyc)22Bri ("Eμ-Myc") mice with C57BL/6JRj background[119] were isolated using negative selection technique (MagniSort Mouse B cell Enrichment Kit, Thermo Fisher Scientific). IgM-positive B-cells were isolated from spleens using a standard antibody cocktail provided in the kit, and IgM-negative B-cells were isolated from bone marrow using the standard antibody cocktail with addition of biotinylated anti-IgM antibody (clone R6-60.2; 1:100; # 553406; BD Biosciences). Tumor cells were isolated from inguinal and axial lymph nodes of male and female tumor-bearing Eμ-Myc transgenic mice of 15 to 52 weeks of age. In brief, single cell suspensions were obtained by mechanical disruption in PBS pH 7.4, filtered and centrifuged for 5 min at 700×g. Supernatants were removed and resuspended in red blood cell lysis buffer (8.3 g/l ammonium chloride in 0.01 M Tris-HCl pH 7.4). Target cells were isolated according to the manufacturer's

protocol. Purity and IgM status of isolated cells were confirmed by flow cytometry (anti-CD19-APC; clone 1D3; 1:1000; # 152410; BioLegend and anti-IgM-PE-Cy7; clone eB121-15F9; 1:100; # 25-5890-82; Thermo Fisher Scientific).

**Periodontal infection model under fatty-acid-enriched diets**. 4-week-old male C57BL/6 mice (University Hospital Jena, Jena, Germany) were randomly divided into groups (n = 6/group) and put on either 16:0- or 18:1-enriched isocaloric diets (20% calories from fat) (Ssniff, Soest, Germany) for a total of 16 weeks. P. gingivalis W50 (# 53978, ATCC) was grown in defined medium, and P. gingivalis/placebo inoculation via oral gavage started at week 10 of the specialized feeding[82]. Animals were sacrificed one week after the final oral infection, and bone/serum samples were collected and homogenized[82] prior to lipid extraction and analysis of PI profiles by UPLC-MS/MS. The experimental protocol was approved by the ethical commission of the University Hospital Jena (UKJ-17-036).

**Studies on planarians**. Planarians used in this work belong to the species Schmidtea mediterranea asexual biotype. Animals were maintained at 19 °C in 1× Montjuïc Salts (1.6 mM NaCl, 1 mM CaCl$_2$, 1 mM MgSO$_4$, 0.1 mM MgCl$_2$, 0.1 mM KCl, 0.1 g/l NaHCO$_3$) and fed with organic veal liver. For RNAi experiments on planarians, templates with T7 promoters appended to both strands[120] were generated for Smed-xbp1, Smed-atf6, and Smed-cct3A. Double-stranded RNA (dsRNA) was synthesized by in vitro transcription with a MEGAscript RNAi kit (Ambion), and dsRNA was injected into the planarian. Following oligonucleotides were used to generate templates for dsRNA production: (1) Smed-xbp1-F: 5′-TAGGTGG-GAATGGTATGGGAAA-3′, Smed-xbp1-R: 5′-CACAACCAAACTCTGACAT TTCG-3′; (2) Smed-atf6-F: 5′-AAGCCAGTTGTTAAGCCAGAAA-3′, Smed-atf6-R: 5′-CCATGATAACCGGGAAATGAAGA-3′; (3) Smed-cct3A-F: 5′-CGTC GTTTTGAGTGGAGTTTTG-3′, Smed-cct3A-R: 5′-TTGATATTGCCATCT CCAATGC-3′[77]. Control animals were injected with gfp dsRNA (GenBank: M62653.1), a sequence not present in the planarian genome. Starved planarians were 7-days starved when starting the RNAi injection while fed planarians were 1-day starved. Planarians were amputated anterior and posterior to the pharynx on day 8 after the first injection and only trunks were processed for lipid extraction. RNA-Seq data from a previous study[77] has been deposited in Expression Omnibus (GEO) with the accession number GSE134013. Significance was determined by q-value (false discovery rate (FDR)) < 0.1 for pairwise comparisons. TPMs are transcripts per million.

**Inhibition of ACC1 and SCD1 by RNA interference**. NIH-3T3 cells (ACC1: $5 \times 10^5/25$ cm$^2$; SCD1: $1.5 \times 10^5/9.5$ cm$^2$) were grown to approximately 60% confluence before being transfected with siRNA duplexes (15 nM) using Lipofectamine RNAiMax transfection reagent (10 µl; Invitrogen). For silencing of Acc1, we used the Acaca (ID 107476) Trilencer-27 Mouse Acc1 siRNA (OriGene, Rockville, MD) that was directed against the sequence 5′-AAGCUACUUUGGUUGAGCAUGGCAT-3′ (84% knockdown efficiency). Control transfections were performed using universal scrambled negative control siRNA duplex (OriGene, Rockville, MD). For SCD1 knockdown, we transfected three different FlexiTube GeneSolution siRNAs (QIAGEN, Hilden, Germany) that targeted the sequences 5′-CACAACAGC TTTAAATAATAA-3′, 5′-TAGTGAGATTTGAATAATTAA-3′, or 5′-CGGTAC AGTATTCTTATAAA-3′, respectively. Medium was replaced after 5 h. On-TargetPlus nontargeting siRNA #1 (Thermo Fisher Scientific) was used as scrambled control.

**Quantitative RT-PCR**. Total RNA was isolated from fibroblasts with E.Z.N.A Total RNA Kit (Omega Bio-tek, Norcross, GA) or innuPREP RNA Mini Kit 2.0 (Analytik Jena, Jena, Germany) and transcribed into cDNA by SuperScript III (Invitrogen) or qScript cDNA Synthesis Kit (Quantabio, Beverly, MA). PCR was performed in Mx3000P 96-well plates using a Stratagene Mx 3005 P qPCR system (Agilent Technologies, Santa Clara, CA) with ≤10 ng/µl cDNA, 1× Maxima SYBR Green/ROX qPCR Master Mix (Thermo Fisher Scientific), and 0.5 µM forward and reverse primer (TIB MOLBIOL, Berlin, Germany) or in Multiply$^\circledR$-µStrip (0.2 ml white) strips (Sarstedt, Nümbrecht, Germany) using a qTower$^3$ G system (Analytik Jena) with 0.75 ng/µl cDNA, innuMIX qPCR DSGreen Standard (Analytik Jena) and 0.6 µM forward and reverse primer (Sigma-Aldrich). Primer information is provided in Supplementary Table 2. PCR conditions using Stratagene Mx 3005 P qPCR system: 95 °C for 10 min followed by 45 cycles of 15 s at 95 °C, 30 s at 61 °C, and 30 s at 72 °C. PCR conditions using qTower$^3$ G: 95 °C for 2 min followed by 60 cycles of 20 s at 95 °C and 45 s at 57 °C. Threshold cycle values were calculated by MXPro – Mx3005P v4.10 software (Agilent Technologies) or qPCRsoft v4.1.3.0 software (Analytik Jena) and normalized to the amount of RNA or Actb.

**Analysis of cell number, viability, and morphology**. Cell number and membrane intactness/viability were determined after trypan blue staining using a Vi-CELL Series Cell Counter (Beckman Coulter, Krefeld, Germany; software: Vi-Cell XR Cell Viability Analyzer, version 2.03 or 2.06.3). Cells were visualized by an Axiovert 200 M microscope with a Plan Neofluar×100/1.30 Oil (DIC III) objective (Carl Zeiss, Jena, Germany). Images were taken using an AxioCam MR3 camera using AxioVision 4.8 (Carl Zeiss).

**Immunofluorescence microscopy.** NIH-3T3 cells ($5 \times 10^3/3.5\ cm^2$) were seeded onto glass coverslips and cultured in presence of vehicle (DMSO), CAY10566 (3 μM), and/or phospholipid vesicles (50 μM) for 42 h. The culture medium, consisting of DMEM plus 10% FCS, was changed against medium additionally containing 16:0 (400 μM), and the treatment with vehicle, CAY10566, and/or phospholipid was continued for 6 h. Cells were fixed with 4% paraformaldehyde (20 min, room temperature), permeabilized using 0.25% Triton X-100 (10 min, 4 °C), and blocked with 5% normal goat serum (Invitrogen, Carlsbad, CA). Samples were incubated with mouse anti-GRP78 antibody (A-10; 1:250; # sc-376768; Santa Cruz Biotechnology, Dallas, TX) overnight at 4 °C and stained with Alexa Fluor 555 goat anti-mouse IgG (1:1000; # A32727; Thermo Fisher Scientific) for 30 min at room temperature. Nuclear DNA was stained by ProLong Diamond Antifade Mountant with DAPI (Thermo Fisher Scientific). Samples were analyzed by an Axiovert 200 M microscope (Carl Zeiss) equipped with a Plan Neofuar×100/1.30 Oil (DIC III) objective (Carl Zeiss). Images were taken with an AxioCam MR3 camera and linearly adjusted in brightness and contrast by AxioVision 4.8 software (Carl Zeiss).

**Annexin-V and propidium iodide staining of apoptotic cells.** NIH-3T3 cells ($3 \times 10^5/9.5\ cm^2$ or $5 \times 10^5/25\ cm^2$) were treated with vehicle (DMSO), CAY10566 (3 μM), etoposide (10 μM), and/or PI vesicles (50 μM) or with vehicle (DMSO), VAL (10 μM), and/or 18:1 (100 μM). After 48 h at 37 °C and 5% $CO_2$, cells were stained with propidium iodide and annexin-V using either an Annexin V Apoptosis Detection Kit APC (Thermo Fisher Scientific) or an Annexin V Apoptosis Detection Kit FITC (Thermo Fisher Scientific) according to the manufacturer's instructions. Cells were analyzed with a BD LSR Fortessa flow cytometer (BD Biosciences), and data were processed by BD FACSDiva (BD Biosciences) and FlowJo (BD Biosciences) or Flowlogic software (Miltenyi Biotech). The gating strategy is outlined in Supplementary Fig. 38.

**Immunoprecipitation.** NIH-3T3 cells ($5 \times 10^5/25\ cm^2$) were treated with vehicle (DMSO) or CAY10566 (3 μM) for 48 h. After washing with ice-cold PBS pH 7.4 and scraping in 500 μl ice-cold lysis buffer (20 mM Tris-HCl pH 7.5, 150 mM NaCl, 1% Triton-X 100, 1 mM EDTA, 1 mM sodium vanadate, 1 mM EGTA, 2.5 mM sodium pyrophosphate, 1 mM β-glycerophosphate, 1 μg/ml leupeptin, 1 mM phenylmethanesulphonyl fluoride), cells were sonicated on ice (3 × 5 s). The lysate was centrifuged ($14,000 \times g$, 10 min, 4 °C), and an aliquot (200 μl, 1 mg/ml total protein) was pre-cleared by incubating with 20 μl of protein A magnetic bead slurry (Cell Signaling, #73778) for 20 min at room temperature. The pre-cleared lysate was separated from the beads using a magnet and incubated with rabbit anti-phospho-tyrosine (p-Tyr-1000, #8954, 1:200) under rotation for 18 h at 4 °C. The formed immunocomplexes were combined with washed protein A magnetic beads (20 μl of slurry). After 20 min at room temperature, beads were collected by magnetic separation and repeatedly washed with lysis buffer. Proteins were detached in SDS/PAGE sample loading buffer (20 μl) and subjected to SDS-PAGE and quantitative proteomics.

**In-gel digestion of proteins.** Protein bands of interest were cut out from the Coomassie-stained SDS-PAGE gels, cut into small pieces, repeatedly washed with 25 mM aqueous $NH_4HCO_3$ and destained with 50% ACN / 25 mM aqueous $NH_4HCO_3$. The proteins were then reduced with 10 mM dithiothreitol at 50 °C for 1 h and alkylated with 55 mM iodacetamide at room temperature in the dark for 45 min. Destained, washed, and dehydrated gel pieces were rehydrated for 60 min in a solution of 12 ng/μl porcine trypsin (Promega) in 25 mM aqueous $NH_4HCO_3$ at 4 °C and incubated overnight at 37 °C. The tryptic peptides were extracted from the gel using 75% ACN / 5% formic acid, and dried down in a vacuum concentrator (SpeedVac, Thermo Fisher Scientific). For nanoUPLC-MS$^E$ analysis samples were reconstructed in 30 μl aqueous 1% formic acid.

**LC-MS$^E$ analysis.** 1 μL of each sample was injected onto an UPLC M-class system (Waters) online coupled to a Synapt G2-si mass spectrometer (Waters). Samples were first on-line pre-concentrated and desalted using a UPLC M-Class Symmetry C18 trap column (100 Å, 180 μm × 20 mm, 5 μm particle size; Waters) at a flow rate of 15 μl/min (0.1% aqueous formic acid). Peptides were eluted onto a ACQUITY UPLC HSS T3 analytical column (100 Å, 75 μm × 200 mm, 1.8 μm particle size; Waters) at a flow rate of 350 nl/min using an increasing acetonitrile gradient with 2-10% B over 5 min, 10–40% B over 40 min, 40–70% B over 7 min, 70-95% B over 3 min, isocratic at 95% B for 2 min, and a return to 1% B (buffers: A, 0.1% formic acid in water; B, 100% acetonitrile in 0.1% formic acid).

The eluted peptides were transferred into the mass spectrometer operated in V-mode with a resolving power of at least 20,000 full width at half height FWHM. All analyses were performed in a positive ESI mode. A solution of 100 fmol/μl human Glu-fibrinopeptide B in 0.1% formic acid/acetonitrile (1:1 v/v) was infused at a flow rate of 1 μl/min through the reference sprayer every 45 seconds to compensate for mass shifts in MS and MS/MS fragmentation mode.

Data were acquired using data-independent acquisition (DIA), referred to as enhanced MS$^E$. MS data were collected using MassLynx v4.1 software (Waters).

**Data processing and protein identification.** The acquired continuum LC-MS$^E$ data were processed using ProteinLynx Global Server (PLGS) version 2.5.2

(Waters) to generate product ion spectra for database searching according to Ion Accounting algorithm[121]. The processed data were searched against Swissprot database (2019_01). The database search was performed at a False Discovery Rate (FDR) of 2% and used stringent criteria. Following search parameters were applied for the minimum numbers of: fragments per peptide (3), peptides per protein (1), fragments per protein (7), and maximum number of missed tryptic cleavage sites (1). Searches were restricted to tryptic peptides with a fixed carbamidomethylation of cysteines.

For quantification, an universal response factor was calculated from trypsin (the averaged intensity of the three most intense peptides)[122].

**Extraction of proteins and lipids for multiomics.** NIH-3T3 cells in 25 cm$^2$ flasks were washed trice with ice-cold PBS pH 7.4 (2 ml) and scraped in methanol (500 μl, −20 °C) and water (500 μl). Samples were combined with chloroform (500 μl, −20 °C) and shaken (1400 cycles/min, 4 °C) for 20 min. After centrifugation ($16,100 \times g$, 4 °C, 5 min), polar and non-polar phases as well as the interphase were collected. Polar and non-polar phases were dried under vacuum at room temperature using an Eppendorf Concentrator Plus system (Hamburg, Germany; polar phase: aqueous application mode; non-polar phase: high vapor pressure application mode). The residue of the non-polar phase was dissolved in methanol and subjected to lipidomics analysis (Supplementary Fig. 3c and 20b). The interphase was washed with methanol (−20 °C) and centrifuged ($16,100 \times g$, 4 °C, 10 min). The pellet was resuspended in denaturation buffer (8 M urea, 100 mM ammonium bicarbonate; 60 μl) and 5-fold diluted in aqueous ammonium bicarbonate (100 mM). After sonication (Branson Ultrasonics™ Sonifier Modell 250 CE, Thermo Fisher Scientific, parameters: 1× 10 s, constant duty cycle, output control: 2, room temperature), extracted proteins were quantified using a Pierce Micro BCA Protein Assay Kit (Thermo Fisher Scientific). Proteins (50 μg in 285 μl) were reduced by dithiothreitol (10 mM) at 55 °C for 30 min and alkylated by iodoacetamide (20 mM, 30 min) at room temperature in the dark. The reaction was stopped by addition of excess dithiothreitol after 30 min. Samples were diluted, subjected to tryptic digestion (16 h, room temperature, sequencing grade modified trypsin; Promega, Madison, WI), acidified with formic acid (4.6 μl), and centrifuged (5 min, 16,000 g, room temperature). Peptides in the supernatant were transferred to Sep-Pak 100 cartridges (30 mg, Waters). After washing with formic acid (1% in water), peptides were eluted with $ACN/H_2O$/formic acid (70/29/1), dried under vacuum, and used for proteomics analysis.

**Quantitative proteomics.** For LC-MS/MS analysis, the dried tryptic peptides were dissolved in 20 μl 0.1% aqueous formic acid. The samples were injected on a nano-ultra pressure liquid chromatography system (Dionex UltiMate 3000 RSLCnano pro flow, Thermo Fisher Scientific) coupled via an electrospray ionization (ESI, nanospray flex ion source) source to an Orbitrap Fusion (Thermo Fisher Scientific). The samples were loaded (20 μl/min) on a trapping column (Acclaim PepMap cartridge, C18, 5 μm, 1 mm × 5 mm, Waters, buffer A: 0.1% formic acid in HPLC-$H_2O$; buffer B: 80% acetonitrile, 0.1% formic acid in HPLC-$H_2O$) with 5% buffer B. After sample loading, the trapping column was washed for 5 min with 5% buffer B (15 μl/min), and the peptides were eluted (300 nl/min) onto the separation column (nanoE MZ PST CSH, 130 Å, C18 1.7 μm, 75 μm × 250 mm, Waters) and separated with a gradient of 5 − 30% B in 120 min. The spray was generated from a steel emitter (Fisher Scientific, Dreieich, Germany) at a capillary voltage of 1900 V. MS/MS measurements were carried out in data dependent acquisition mode (DDA) using a normalized HCD collision energy of 30% in full speed mode. Every second a MS scan was performed over an m/z range from 350-1600, with a resolution of 120,000 at m/z 200 (maximum injection time = 120 ms, AGC target= 2e5). MS/MS spectra were recorded in the ion trap (rapid scan mode, maximum injection time= 60 ms, maximum AGC target = 1e4, intensity threshold: 1e5, first m/z: 120), a quadrupole isolation width of 1.6 Da and an exclusion time of 60 seconds.

For data analysis, raw files were analyzed with ProteomeDiscoverer 2.4 (Thermo Fisher Scientific). For peptide and protein identification, the LC-MS/MS were searched with SequesHT against a mouse database (SwissProt, 17,023 entries) and a contaminant database (116 entries). The following parameters were used for the database search: mass tolerance MS1: 6 ppm, mass tolerance MS2: 0.5 Da, fixed modification: carbamidomethylation (Cystein), variable modification: Oxidation (Methionine), variable modification at protein N-terminus: Acetylation, Methionine loss, Methionine loss + Acetylation. Pathway analysis (p38 MAPK signaling, UPR, autophagy, programmed cell death, PI biosynthesis/metabolism) was performed based on proteins listed in Reactome-, Wiki-, and Kegg-pathways (Supplementary Data 2–6). Moreover, we extracted pathway-related proteins from recent overview articles[41,123–128].

Percolator were used for FDR calculation. For feature detection, Minora Feature Detection was used with default settings. For label free quantification, the Precursor Ions Quantifier was used with the following parameters: Peptides to use: unique peptides, Precursor Abundance Based On: Area, Minimum Replicate Features: 100%, Normalization Mode: Total Peptide Amount, Protein Abundance Calculation: Summed Abundances, Top N: 3. Data were further processed using RStudio (version 1.4.1106). For quantitative comparison, the reported protein intensities were used. Volcano plots were generated with the rstatix R package using an unpaired, two-tailed Welch t test and Benjamini–Hochberg correction to calculate adjusted p values or with GraphPad Prism 9.0 (GraphPad Software, San Diego, CA) using an unpaired, two-tailed multiple t test and Benjamini, Krieger,

and Yekutieli correction (false discovery rate: 0.05). Heatmaps and z-scores were generated using GraphPad Prism 9.0 or the ComplexHeatmap R package (cluster_rows: proteins, cluster_number: 15).

**Incorporation of phospholipids into fibroblasts**. Phospholipids (50 μM, each) were suspended in DMEM containing 10% FCS (except for serum depletion), vigorously mixed, and sonicated at 50 °C for 20 min to form phospholipid vesicles, which were then supplemented to the cell culture medium. When combined with siRNA treatment, medium containing siRNA/lipofectamine RNAiMax complex was exchanged against phospholipid-containing medium after 5 h.

Alternatively, PI(18:1/18:1) was instantly incorporated into fibroblasts using the Fuse-It-L membrane fusion system (Ibidi, Martinsried, Germany). In brief, PI(18:1/18:1) (10 nmol), dissolved in chloroform, was added to the lyophilized Fuse-It-L reagent and thoroughly mixed. The solvent was evaporated, and the complex was resuspended in 20 mM HEPES pH 7.4 (25 μl) to obtain the fusogenic mixture that was sonified for 15 min below 25 °C. An aliquot (5-8 μl) was diluted in PBS pH 7.4 (0.5 ml) and transferred to fibroblasts ($3.5 \times 10^5$ / well of a 6-well plate), whose culture medium has been removed. After 5 min at 37 °C, the fusogenic mixture was again changed against cell culture medium. Since strong differences in lipid uptake were observed between different batches of the kit, we controlled the incorporation of PI(18:1/18:1) for each dataset by UPLC-MS/MS. Successful lipid uptake was defined as ≥30% increase of cellular PI(18:1/18:1) ratios.

**Co-regulated lipid networks**. Lipid co-regulation was defined as Pearson correlation values >0.7 and is visualized in a random co-regulation network implemented in Cytoscape 3.3 (Cytoscape Consortium)[129]. Networks were calculated from mean cellular lipid proportions and were correlated with mean phospho-p38 MAPK levels. Negative correlation of lipid species with phospho-p38 MAPK levels is highlighted for Pearson correlation values < −0.6. Pearson values were calculated using Microsoft Excel 2016 (Microsoft Office Professional Plus 2016, Microsoft, Redmond, WA). Network nodes indicate individual lipid species and edges show co-regulations above the threshold.

**Data analysis and statistics**. Data are expressed as mean ± s.e.m. of $n$ independent experiments. Samples were not blinded, and sample size was not predetermined by statistical methods. Shapiro-Wilk tests were used to investigate the data with similar variance between groups for normal distribution. Non-transformed or logarithmized data was statistically evaluated by one-way or two-way ANOVAs for independent or correlated samples followed by Tukey HSD post hoc tests or by two-tailed student $t$ test for paired or unpaired samples using a two-sided α level of 0.05. $P$ values <0.05 were considered statistically significant. Outliers were determined using a Grubb's test. Data were analyzed using Microsoft Excel 2016 (Microsoft Office Professional Plus 2016, Microsoft), and statistical calculations were performed using SigmaPlot 13 and 14 (Systat Software GmbH, San Jose, CA), GraphPad InStat 3.10, GraphPad Prism 8.0, or GraphPad Prism 9.0 (GraphPad Software). Heatmaps were created using Morpheus (https://software.broadinstitute.org/morpheus) or GraphPad Prism 9.0 (GraphPad Software) from relative or absolute intensities that were normalized to control. Principal component analysis was performed with Origin 2020 (OriginLab, Northampton, MA).

**Reporting summary**. Further information on research design is available in the Nature Research Reporting Summary linked to this article.

## Data availability

Source data are provided with this paper.

The mass spectrometry lipidomics data generated in this study have been deposited in the Metabolomics Workbench database (an international repository for metabolomics data and metadata, metabolite standards, protocols, tutorials and training, and analysis tools[130]) under accession code ST001740 [131]. The mass spectrometry proteomics data have been deposited to the ProteomeXchange Consortium via the PRIDE[132] partner repository with the dataset identifier PXD025396 and PXD031890. All other data generated or analyzed during this study are provided in this published article, the Source Data, Supplementary Information, or Supplementary Data.

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

## Acknowledgements

We thank Felix Benscheidt, Yvonne Hupfer, Vajiheh Jafari, Viktoria Iffarth, Martin Niebergall, Jinghong Peng, Katrin Schubert, Vanessa Sorge, Franziska Stanzel, and Maria Völkel for technical assistance in performing experimental methods and Simona Pace for consultation in the design of mouse studies.

A.K. was supported by the German Research Council (GRK 1715 and KO 4589/4-1), the Phospholipid Research Center (AKO-2022-100/2-2, AKO-2019-070/2-1, AKO-2015-037/1-1), the University of Jena (DRM/2013-05 and 2.7-05), and a Strategy and Innovation Grant from the Free State of Thuringia (41-5507-2016) and the Leibniz ScienceCampus Infec-toOptics (SAS-2015-HKI-LWC). R.W. was supported by the Carl Zeiss foundation. C.G.-E. was funded by the Leibniz Institute on Aging-Fritz Lipmann Institute (FLI) and currently holds a Maria Zambrano fellowship (call for requalification of the Spanish University System 2021-2023) at the University of Barcelona. The FLI is a member of the Leibniz Association and is financially supported by the Federal Government of Germany and the State of Thuringia. US-S was supported by the Interdisciplinary Center for Clinical Research of the University Hospital Jena (IZKF UKJ FF02) and the Federal Ministry of Education and Research (01EC1901B, Project 2). K.T. acknowledges support from the MESI-STRAT project (grant agreement 754688), the PoLiMeR Innovative Training Network (Marie Skłodowska-Curie grant agreement 812616) and the ARDRE COFUND Training Network (Marie Skłodowska-Curie grant agreement No 847681) which all received funding from the European Union Horizon 2020 Research and Innovation Program; the German Tuberous Sclerosis Foundation and Stichting TSC Fonds. M.K. was supported by the University of Innsbruck (project no: 316826) and the Tyrolian Research Fund (project no: 18903).

## Author contributions

A.K. designed and M.T., A.G., H.P., E.G., L.W., K.L., and J.G. performed and analyzed cell-based experiments and conducted lipidomic studies. N.W., M.T., and A.S. analyzed the composition of immunoprecipitates by quantitative proteomics. F.T. and A.K. designed and F.T., M.T., A.G., and K.N. performed experiments on murine peritoneal macrophages, murine brain, and mice with defective *Scd1* mutation. K.L., R.W., and C.K. provided B cell lymphoma and performed immunoblotting on respective cell populations. L.R. prepared hematopoietic stem cells and myeloid progenitor cells from young and old mice. A.D. and U.S. designed and performed experiments related to *P. gingivalis* infection. M.M., M.A., and J.P. generated the sorafenib-resistent hepatocarcinoma cell line. C.G-E and O.G-G housed and treated planarians, performed knockdown experiments on these animals, and provided transcriptome data. H.S. and K.T. provided methodology for quantitative proteomics. A.G., M.H., S.H., and M.K. performed, and A.G., M.H., M.K., and A.K. analyzed the data from quantitative proteomics. A.K. conceived the project, and A.K. and M.T. wrote the manuscript, which was edited and approved by all authors.

## Competing interests

The authors declare no competing interests.
