## [Peer Review File · Nature Communications]

PI(18:1/18:1) is a SCD1-derived lipokine that limits stress signalingReviewers' Comments:

Reviewer #1:

Remarks to the Author:

In their paper "PI is a SCD1-derived lipokine that limits stress signaling" Maria Thuermer et al suggested that PI(18:1/18:1) is SCD1-derived lipokine, which maintains cell homeostasis, morphology and survival by suppressing p38 MAPK stress signaling and limiting stress responses, including the UPR and autophagy.

Authors presented a lot of data confirming, that MUFA containing PI play some function in signaling, but it seems, that author's conclusion is too strong. Presented results demonstrated, that stress conditions decrease activity of stearoyl-CoA desaturase (SCD) and content of MUFA PI suggesting another explanation of authors results. One can speculate, that inhibition of p38 kinase and prevention of the development of UPR and autophagy occur in normal condition, when PI(18:1/18:1) present in relatively high concentrations. Stress conditions activate p38 kinase induced UPR and autophagy and lead to inhibition of SCD1 and decrease in amount of MUFA-PI, promoting further activation of p38 kinase, UPR, autophagy and worsening stress consequences.

I have several concerns after reading a manuscript.

1. From results of experiments to induce programmed cell death in fibroblasts authors made conclusion, that the cellular proportion of PI(18:1/18:1) decreases in (pre)apoptotic cells and the depletion of this lipid is accompanied by the induction of p38 MAPK stress signaling across diverse cell lines. Not all presented results supported this conclusion.

i) Presented results demonstrated, that not all inducers produced inverse correlation between MUFA PI content and activation of p38. For instance, staurosporine decreased MUFA PI content, but didn't induce any changes in p38 in all studied time points. Etoposide didn't produce any changes in MUFA-PI after 24h of incubation, but increase p38 activity. Serum decreased MUFA PI concentration with no significant changes in p38 activity.

ii) Treatment of NIH-3T3 fibroblasts with VAL increased p38 activity after 6, 24, 48h incubation, but decreased MUFA PI content only in 48 h, while increased in early time points (6, 24h).

iii) Addition of VAL to some cell lines induced inverse correlation between MUFA PI content and activity of p38 but these phenomenon was found not in all studied cell lines (Fig. 2e).

I agree with author's statement, that heterogeneity in answers is not surprising in light of the variable connectivity of the p38 MAPK signaling network for different cell types, but for now it looks like inverse correlation between MUFS PI content and activity of p38 is not universal principle of cell response to stress conditions.

Another concern about these experiments is incubation time. Authors find best inverse correlations between MUFA PI and p38 activity in cells incubated for 48h. They also demonstrated, that many cytotoxic agents substantially decreased Scd1 mRNA levels between 6 to 48 h (mostly-24, 48 h) and SCD1 protein expression in 48 h. One can conclude from analysis of supplementary fig.18e that cell death in 48h was ~50%. It looks like too late. One can speculate, that correlation between MUFA PI and p38 activity start developing when cell already decided to die and is a result of postmortem changes, which have nothing to do with signaling in alive cells. Data that addition of exogenous PI did not restore viable cell numbers (Supplementary Fig. 18) can serve as indirect evidence of this suggestion.

2. On P6 authors wrote: " We ... combined the data in a co-regulated phospholipid network (Fig. 1a). Positively correlated phospholipids are located in close proximity and interconnected, whereas non- or counter-regulated phospholipids form separate clusters". It is not clear how this network was built.

3. Results of experiments to assess impact of CAY10566 on protein tyrosine phosphorylation looks confusing. Western blotting demonstrated, that treatment with CAY10566 increased recognition by

anti-phospho tyrosine antibody of proteins with molecular weight ~100kDa, 70kDa and 35kDa, while staining of protein with 55 kDa decreased (Fig. 5a). At the same time only 55kDa protein was isolated by immunoprecipitation using antibodies against phosphorylated tyrosine (Fig. 5b). Why proteins with molecular weights 100 kDa, 70 kDa and 35 kDa was not isolated by immunoprecipitation?

4. To investigate whether SCD1-derived phospholipids counteract stress signaling authors inhibited SCD1 in fibroblasts by CAY10566 and then added different exogenous PI. Obtained results demonstrated, that cells treated with CAY have ~90% of viability, while viability of cells incubated with CAY and treated with exogenous MUFA PI, is ~75% (fig 7e, suppl fig 8b,). How authors can explain decrease of cell viability after addition of MUFA PI, which supposed to protect cells?

5. Initial product of SCD is non-esterified MUFAs, which than converted into MUFA PI. The cytotoxic settings inhibit SCD activity, markedly decrease not only content of MUFA PI but also decrease amount of free MUFAs and increase the amount of free SFAs (Fig. 3b). It can be suggested, that biological effects of SCD inhibition can be at least partially dependent on changes in ratio free MUFA/free SFA. Treatment of cell with inhibited SCD with free MUFA/SFA can serve as useful control for experiments in which different exogenous PI were added to cells.

6. In support of their hypothesis, authors studied level of PI(18:1/18:1) in different cells, tissues, and organisms being in various physiological states characterized by different stress tolerance such as tumorigenesis, drug-resistance, starvation, aging, and infection. They assess just the level of PI(18:1/18:1) and find both increase and decrease in the amount of this lipid. It looks like without assessment of other players, such as activity of p38, level of UPR and autophagy these results can provide only circumstantial evidence of the role of PI (18:1/18:1) if any.

Reviewer #2:

Remarks to the Author:

The manuscript of Thuermer et al. discusses PI(18:1/18:1) as stress lipokine and biomarker.

General remarks: this is a very extensive manuscript and I have to advice condensing it.

A paper by Pein et al is cited at multiple instances as in preparation. This is inappropriate as I cannot access this information.

Indeed SCD1 does produce 18:1-CoA and further as the authors correctly outline "PI(18:1/18:1) is biosynthesized from SCD1-derived 18:1-CoA". However, in my opinion there is a missing puzzle piece here, PIS Phosphatidylinositol synthase. If I am not mistaken is this enzyme ultimately catalyzing the production of PI18:1. However, it seems that it has been left out of the investigation. While I agree that PI18:1 might be a promising stress biomarker is the presented mechanistic evidence on how this exactly works rather descriptive in my view. We know now that SCD/PI18:1/p-MAPK (etc...) are connected but how exactly? Does PI18:1 block expression on mRNA level? From F7 and F8 it seems as if phosphorylation of MAPK is affected, as mentioned below also I am not a proteomics person, but I wonder how quantitative such a finding is based on western plot analysis. In other words, this manuscript clearly establishes a link between SCD/PI18:1/MAPK but in my view comes short in explaining how these three are now exactly connected. Lots of interesting observations are presented underlining PI18:1 as possibly useful biomarker, however the causal connections are not sufficiently explored.

As this is a very extensive manuscript I would like to structure this review along the presented figures.

Figure 1:

What is the added value of 1a? Network analysis has become fancy, but I sometimes question myself what does it add?

Could the heatmaps in 1c be clustered? As in all 18:1 lipids in one cluster than this becomes visually a lot clearer.

The ultimate point of F1 is that cytotoxic conditions decrease MUFA, this is principally a sound conclusion from the provided data. However, what about the free fatty acid and LPI fractions? This might also be an effect of PLA releasing MUFA.

Figure 2:

The objective of this paragraph is to correlate PI18:1 with MAPK.

Indeed, for some of the tested cell lines there is a correlation between MAPK and PI18:1.

Figure 3:

A – this should be presented as volcano plot

This paragraph shows that there is a correlation between SCD and MUFA/PUFA/SFA fractions. The conclusions here seem sound.

Figure 4:

Panel 4d is misleading, up to now the authors showed MUFA-PI changes as a fraction of total, now suddenly the panel focuses on a specific ratio. How come? Why is this suddenly introduced? The siRNA experiment is crucial. These inhibitors do a thousand things plus inhibiting SCD so I would not be surprised if CAY10566 induces cell death/stress to a certain extent. Along these lines, did CAY10566 also change other lipid fractions? In my view panel E is the most relevant as it shows a correlation between MAPK and SCD knock down.

Figure 5:

I am not a proteomics person, but 1-2 matched peptides are in my view not enough for a solid protein identification. Hence, the data in this panel looks quite shaky to me. Please comment. Moreover, a PLA response would be quite logic under stress conditions, this would correlate with panel 4i where CAY shows a clear trend of inducing apoptosis, please comment.

Figure 6:

Subsequently the authors seek to confirm their in vitro findings in an in vivo model of SCD deficient mice. The mice showed the expected decreased MUFA levels as well as increased MAPK levels etc. Panel 6E I would show as volcano plot.

Figure 7:

As a next step the authors carry out an “add-back” experiment using PI liposomes.

How relevant are the employed 50microM concentrations? How does this relate to the in vitro/in vivo concentration of PI18:1?

The rest of this paragraph is clear

Figure 8:

How do the authors explain the effects of CAY here? In other words, PI18:1 is a SCD derived lipokine which limits stress response (MAPK, URP etc). Now, we add PI18:1 and it blocks p-p38MAPK etc. But when we add PI18:1 together with CAY the effects are absent? Maybe I misunderstand the setup, but CAY mainly blocks SCD which produces PI18:1. However, in this setup we add PI18:1 so SCD is not needed to produce PI18:1. In turn, how does CAY diminish the effects of PI18:1 in this setup?

Figure 9:

This paragraph is clear, the authors investigate PI18:1 as possible stress biomarker in several systems.

Figure 10:

This figure summarizes the findings. As I mentioned above, in my view PIS is missing here and has not been mentioned throughout the manuscript.

Reviewer #3:

Remarks to the Author:

The manuscript by Thuermer et al. "PI (18:1/18:1) is a SCD-1 derived lipokine that limits stress signaling presents the data that PI (18:1/18:1) , an SCD-1 generated lipid inhibiting p38 MAPK phosphorylation has a strong inhibitory effect on the stress response. The authors thoroughly characterized the effects of the SCD1 inhibition on lipid composition and on stress signaling in various contexts, however, they do not reach deeper into the molecular mechanism explaining the action of PI (18:1/18:1). The topic is appropriate and should garner attention of the broad audience. Aside from the lack of mechanistic explanation there are a few issues that need attention should the manuscript be considered for publication.

Detailed comments:

1. The effect of SCD1 inhibition on protein tyrosine phosphorylation. The data in figure 5 are not very convincing. Why did the authors quantify the band showing on Western blot around 100kDa and indeed different between CAY and W/o samples, and then decided to look at two bands of completely different mass that were different on the Coomassie stained gel (which obviously doesn't show only tyrosine phosphorylation but total protein) and present tyrosine phosphorylated peptides which are NOT the dominant protein in these bands? Were the bands close to 100kDa also analyzed? If anything, the whole samples should be analyzed for differential tyrosine phosphorylation using pY-IP. Then these results would be meaningful. Also, a table showing the processed data should be added as a supplemental table (aside from the raw data submitted to the database).
2. Why was only PI(18:1/18:1) chosen for further investigation? PI(16:0/16:0) had even stronger effect on the BiP levels, this should be at least discussed.
3. (minor) some persistent spelling errors should be fixed, for example, principal, not principle components (pages 11, 18) and Coomassie, not coomaissie (figure 5 legend).

Response to the editor and reviewers

We are very grateful for the helpful comments by the editor and reviewers, which helped us to significantly improve the quality of our manuscript. In the following, we provide a detailed point-by-point list of the changes made (“answer”) in response to the editor’s instructions and referees’ suggestions (“comment”).

REVIEWER #1

Comment: In their paper “PI is a SCD1-derived lipokine that limits stress signaling” Maria Thuermer et al suggested that PI(18:1/18:1) is SCD1-derived lipokine, which maintains cell homeostasis, morphology and survival by suppressing p38 MAPK stress signaling and limiting stress responses, including the UPR and autophagy. Authors presented a lot of data confirming, that MUFA containing PI play some function in signaling, but it seems, that author’s conclusion is too strong. Presented results demonstrated, that stress conditions decrease activity of stearoyl-CoA desaturase (SCD) and content of MUFA PI suggesting another explanation of authors results. One can speculate, that inhibition of p38 kinase and prevention of the development of UPR and autophagy occur in normal condition, when PI(18:1/18:1) present in relatively high concentrations. Stress conditions activate p38 kinase induced UPR and autophagy and lead to inhibition of SCD1 and decrease in amount of MUFA-PI, promoting further activation of p38 kinase, UPR, autophagy and worsening stress consequences.

Answer: We thank the reviewer for this point which is well taken. Our kinetic data on p38 MAPK activation (Fig. 2b) and PI(18:1/18:1) depletion (Fig. 1c) supports the hypothesis that cytotoxic stress triggers an early stress response (within 10 min to 6 h), which precedes the drop in SCD-1/PI(18:1/18:1) and p38 MAPK, UPR, and autophagy activation (at 6 to 48 h). While our data clearly show that PI(18:1/18:1) interferes with stress signaling (p38 MAPK, UPR, and autophagy), the directionality and dependency of the stress signaling events are less defined, as now mentioned on page 33, lines 13-15. To obtain further mechanistic insights, we performed quantitative proteomics studies on valinomycin or myrtucommulone A-treated fibroblasts, compared the proteomic changes to those from selective SCD1 inhibition, and explored the potential of PI(18:1/18:1) to compensate for the effects (new Fig. 8 and Supplementary Figs. 23 and 24, page 23-25). Based on the results, we developed hypotheses on how SCD1-derived PI(18:1/18:1) mediate its stress-reducing activity, as now described on page 24, line 14 to page 25, line 18 and in Supplementary Note 7 and discussed on page 34, line 6 to page 36, line

17. In particular, we identified the catalytic subunit of protein phosphatase 2A as SCD1/PI(18:1/18:1)- regulated protein that depletes under cytotoxic stress (Fig. 8a-c). This finding is of particular interest considering that protein phosphatase 2A dephosphorylates major players in p38 MAPK activation (PMID: 15569672, 18299321, 16407301), the UPR (PMID: 32457508), and autophagy (PMID: 26310906). Whether early (p38 MAPK-related) stress signaling initiates SCD1 depletion and thus aggravates stress conditions or whether independent mechanisms (triggered by cytotoxic stress) are causative, cannot be fully answered. We now discuss these important aspects on page 32, lines 11-15

and page 33, lines 7-18. Moreover, we carefully revised the abstract (page 3) and introduction (page 5, line 23 and page 6, lines 1-7) to avoid the impression of overstating our findings.

Comment: I have several concerns after reading a manuscript. 1. From results of experiments to induce programmed cell death in fibroblasts authors made conclusion, that the cellular proportion of PI(18:1/18:1) decreases in (pre)apoptotic cells and the depletion of this lipid is accompanied by the induction of p38 MAPK stress signaling across diverse cell lines. Not all presented results supported this conclusion. i) Presented results demonstrated, that not all inducers produced inverse correlation between MUFA PI content and activation of p38. For instance, staurosporine decreased MUFA PI content, but didn't induce any changes in p38 in all studied time points. Etoposide didn't produce any changes in MUFA-PI after 24h of incubation, but increase p38 activity. Serum decreased MUFA PI concentration with no significant changes in p38 activity. ii) Treatment of NIH-3T3 fibroblasts with VAL increased p38 activity after 6, 24, 48h incubation, but decreased MUFA PI content only in 48 h, while increased in early time points (6, 24h).

Answer: We agree that this point requires more intense discussion, thereby considering biphasic kinetics as suggested by the reviewer and outlined in our answer to comment 1. Diverse cytotoxic stressors rapidly activate p38 MAPK with varying mechanisms, kinetics and magnitudes, and we assume that this first heterogenic phase of p38 MAPK activation takes place within 10 min to 6 h (phase I) in our experimental system. The second phase of p38 MAPK activation from 6 h to 48 h seems instead to be more consistent under our cytotoxic conditions and driven by the depletion of SCD1 and PI(18:1/18:1). Our conclusions on a common mechanism exclusively refer to the second phase of p38 MAPK activation, for which we largely obtained good correlations with only few exceptions that we cannot explain, as now described on page 33, lines 9-18. Critical points addressed by the reviewer are discussed in the following and summarized in the new Supplementary Note 10.

Valinomycin: p38 MAPK phosphorylation following treatment with valinomycin significantly increased up to 6 h (3.4-fold increase, $P = 0.04$; phase I) and then decreased again at 24 h by trend (2.8-fold increase) before strongly and significantly rising again at 48 h (4.1-fold increase, phase II). The second phase of p38 MAPK activation correlates with a decrease in PI(18:1/18:1) levels.

Etoposide non-significantly decreased PI(18:1/18:1) levels (17%, $p = 0.99$) and non-significantly increased p38 MAPK phosphorylation (3.3-fold, $p = 0.09$) at 24 h. While the depletion of PI(18:1/18:1) is less pronounced as we would expect from p38 MAPK phosphorylation, interpretations should be made with caution considering the non-significant effects at this time point. Interestingly, etoposide is the only cytotoxic stressor that did not evoke p38 MAPK activation within 6 h, and it is tempting to speculate that etoposide shows a delayed phase I response that might overlap at 24 h with the phase II activation of p38 MAPK.

Serum depletion decreased PI(18:1/18:1) levels by 78% and increased p38 MAPK phosphorylation 2.1-fold at 48 h, which is relatively low when compared to other cytotoxic stressors. Moreover, the effect on p38 MAPK phosphorylation is not significant when applying a mixed-effects model (REML) + Tukey HSD *post hoc* tests of log data across all cytotoxic conditions. However, p38 MAPK activation is significant, when directly comparing serum depletion with control (see Fig. X2).

Fig. X2 Activation of p38 MAPK by serum depletion (Ser) of fibroblasts for 48 h. Phosphorylation of p38 MAPK. Paired data from $n = 5$ independent experiments. P values given vs. vehicle control; two tailed paired t -test of log data.

Considering that p38 MAPK activation is an ATP-dependent process, one possible explanation for the poor activation of p38 MAPK might be the low energy/ATP status of fibroblasts upon serum depletion, as indicated by the substantial (and highly significant) increase of the ADP/ATP ratio (Fig. X3). These data is part of a manuscript, which is in preparation for resubmission to *Nature Communications* (Pein et al., 2022). Please find the manuscript among the uploaded files.

Fig. X3 Cell death lowers the cellular energy status. Fibroblasts were cultivated under diverse cytotoxic conditions and the cellular ADP/ATP ratio was determined. Mean + s.e.m. and single data from $n = 3$ independent experiments. * $P < 0.05$, *** $P < 0.001$ vs. vehicle control; repeated measures one-way ANOVA + Tukey HSD *post hoc* tests.

Staurosporine decreased PI(18:1/18:1) levels and strongly induced p38 MAPK phosphorylation. However, we decided to grey out these results because the compound is a pan-kinase inhibitor and hyperphosphorylation of p38 MAPK might therefore not necessarily indicate an activation. The time- dependent phosphorylation of p38 MAPK by staurosporine is now shown in the source data file for Fig. 2b and the grey color in the heatmap has been defined in the figure legend.

Comment: iii) Addition of VAL to some cell lines induced inverse correlation between MUFA PI content and activity of p38 but these phenomenon was found not in all studied cell lines (Fig. 2e). I agree with author's statement, that heterogeneity in answers is not surprising in light of the variable connectivity of the p38 MAPK signaling network for different cell types, but for now it looks like inverse correlation between MUFS PI content and activity of p38 is not universal principle of cell response to stress conditions.

Answer: Thank you for addressing this point, which we now discuss in more detail in Supplementary Note 1. The transfer of the experimental conditions (which were optimized for fibroblasts) to other cell lines did indeed not always induce inverse correlations between the proportion of MUFA-PI and p38 MAPK activation. Inverse correlations were observed for MCF-7, HEK-293, monocytes, and HepG2 cells (either reaching significance or by trend), whereas neither MUFA-PI levels nor p38 MAPK phosphorylation were affected in MM6, HeLa, and, given the variance, potentially also HT29 cells. While we cannot exclude that distinct cell types do not engage the here reported mechanism, we rather consider the different responsiveness to valinomycin being critical for the differences. Note that always the same valinomycin concentration has been used independent of the sensitivity of the cell lines/types towards the cytotoxic drug. As expected from such an experimental design, i) a strong reduction of PI(18:1/18:1) was associated with a strong increase of p38 MAPK phosphorylation (MCF-7, HepG2), ii) a moderate reduction of PI(18:1/18:1) by trend correlated with a moderate increase of p38 MAPK activation again by trend (HEK293, monocytes), and ii) cell lines, whose PI(18:1/18:1) levels were non-responsive to valinomycin (at the applied concentration) did also not show p38 MAPK activation (MM6, HeLa, and potentially HT29). There is one exception, which we cannot readily explain, namely HUVEC, for which we observed both a decrease of PI(18:1/18:1) ratios and p38 MAPK phosphorylation, though neither of the two effects became significant, as now mentioned in Supplementary Note 1. Together, our findings show a consistent correlation between PI(18:1/18:1) levels and p38 MAPK phosphorylation for most

cell types/lines but do not answer the question

whether the PI(18:1/18:1)-dependent regulation of p38 MAPK is universal (for optimized concentrations and incubation times across cell lines) or limited to distinct cell lines/cell types. The conclusions that can be drawn from this experiment are now clearly expressed in Supplementary Note 1.

Comment: Another concern about these experiments is incubation time. Authors find best inverse correlations between MUFA PI and p38 activity in cells incubated for 48h. They also demonstrated, that many cytotoxic agents substantially decreased Scd1 mRNA levels between 6 to 48 h (mostly-24, 48 h) and SCD1 protein expression in 48 h. One can conclude from analysis of supplementary fig.18e that cell death in 48h was ~50%. It looks like too late. One can speculate, that correlation between MUFA PI and p38 activity start developing when cell already decided to die and is a result of postmortem changes, which have nothing to do with signaling in alive cells. Data that addition of exogenous PI did not restore viable cell numbers (Supplementary Fig. 18) can serve as indirect evidence of this suggestion.

Answer: The reviewer highlights a very critical point when performing mechanistic studies on cell death. We carefully considered this challenge when designing the experiments and therefore decided to perform all studies (lipidomics, proteomics, Western Blot, etc.) that do not determine cell death phenotypes (cell number, PI/annexin staining, etc.) only on cells that maintained the capability to still attach to the cell culture flasks. By intensive washing before detachment and harvesting of cells, we removed dead, detached cells and probably to some degree also rounded-up late apoptotic cells that are only loosely attached. Accordingly, the former Supplementary Fig. 18e shows total cell numbers, whereas the former Supplementary Fig. 18a-d (now Supplementary Fig. 25) exclusively refers to (assumably viable) attached cells. We now mention this experimental design more prominently on page 7 (line 16) and not only in the method section (page 40, line 23 to page 41, line 3). In support of our assumption that the attached cells are still mostly viable, we observed specific metabolic and signaling pathways to be switched on or off (e.g., p38 MAPK activation versus SCD-1 expression). Moreover, quantitative proteomics studies on valinomycin- or myrtucommulone-treated or serum depleted fibroblasts revealed both up- and down-regulation of proteins, as shown in the new Supplementary Fig. 23a, b and e and Supplementary Data 1. To further explore why supplementation of PI(18:1/18:1) failed to prevent the decrease in cell numbers under cytotoxic conditions, we co- treated fibroblasts with valinomycin and free 18:1, which is efficiently incorporated into PI and other phospholipids (new Supplementary Fig. 21a). Again, neither viable cell numbers nor membrane integrity were reduced by 18:1 (new Supplementary Fig. 27a, b). We then performed PI/annexin V staining on the fraction of viable (attached) cells and found that 18:1 effectively reduces the proportion of necrotic/apoptotic cells (new Supplementary Fig. 27c, d). Together, supplementation of SCD1 products (18:1 and/or PI(18:1/18:1)) seems to be beneficial for the vitality of surviving cells under cytotoxic stress, whereas overall cell death induction or proliferation are not substantially influenced by 18:1 under our experimental conditions. These conclusions are now explicitly stated on page 27, lines 5-15.

Comment: 2. On P6 authors wrote:” We ... combined the data in a co-regulated phospholipid network (Fig. 1a). Positively correlated phospholipids are located in close proximity and interconnected, whereas non- or counter-regulated phospholipids form separate clusters”. It is not clear how this network was built.

Answer: Done, we further extended our description on how co-regulated lipid networks were calculated and moved this description from the “Data analysis and statistics” section in the Methods to a new section entitled “Co-regulated lipid networks” (page 57-58).

Comment: 3. Results of experiments to assess impact of CAY10566 on protein tyrosine phosphorylation looks confusing. Western blotting demonstrated, that treatment with CAY10566 increased recognition by anti-phospho tyrosine antibody of proteins with molecular weight

~100kDa,70kDa and 35kDa, while staining of protein with 55 kDa decreased (Fig. 5a). At the same time

only 55kDa protein was isolated by immunoprecipitation using antibodies against phosphorylated tyrosine (Fig. 5b). Why proteins with molecular weights 100 kDa, 70 kDa and 35 kDa was not isolated by immunoprecipitation?

Answer: Agreed, we deleted panel A to avoid confusion and moved the former Figure 5 to the supplementary information (Supplementary Fig. 12). We assume that immunoprecipitation of phospho-tyrosine proteins results in a substantial pull-down of proteins that either bind to phospho- tyrosine motives or other domains of tyrosine-phosphorylated proteins. We further suppose that these proteins dominate the staining of the polyacrylamide gel. Thus, the proteins at 35, 70, and 100 kDa, which are phosphorylated at tyrosine and regulated by CAY10566, might be present in the immunoprecipitate, despite not showing visible bands, when assuming that their concentrations are considerably lower as compared to other (not tyrosine-phosphorylated) proteins. Since our proteomics approach will hardly be able to unambiguously identify these minor proteins, we discontinued respective studies and focused instead on Coomassie-stained bands with clearly different intensities between groups.

Comment: 4. To investigate whether SCD1-derived phospholipids counteract stress signaling authors inhibited SCD1 in fibroblasts by CAY10566 and then added different exogenous PI. Obtained results demonstrated, that cells treated with CAY have ~90% of viability, while viability of cells incubated with CAY and treated with exogenous MUFA PI, is ~75% (fig 7e, suppl fig 8b,). How authors can explain decrease of cell viability after addition of MUFA PI, which supposed to protect cells?

Answer: Excellent point, which we now discuss on page 27, line 1-15. CAY10566 reduced the proportion of viable non-necrotic/non-apoptotic cells from 98 to 87% (all cells analyzed including detached cells), and co-treatment with PI(18:1/18:1) resulted in a further decrease to 82%. Statistical analysis implies that the ratio of viable cells does not substantially differ between CAY10566- and CAY10566/PI(18:1/18:1)-treated cells ($p = 0.93$, repeated measures ANOVA + Tukey *posthoc* test). More intriguing seems to be that PI(18:1/18:1) does not increase the proportion of viable cells, which is in line with our findings on fibroblasts, where PI(18:1/18:1) or 18:1 supplementation failed to attenuate the cytotoxic stress-induced decrease in cell numbers (Supplementary Fig. 25e and 27a). It is therefore tempting to speculate that 18:1 is beneficial only for the still attached cells that survive SCD1 inhibition, as now shown for 18:1-supplemented fibroblasts under cytotoxic conditions (new Supplementary Fig. 28c, d).

Comment: 5. Initial product of SCD is non-esterified MUFAs, which than converted into MUFA PI. The cytotoxic settings inhibit SCD activity, markedly decrease not only content of MUFA PI but also decrease amount of free MUFAs and increase the amount of free SFAs (Fig. 3b). It can be suggested, that biological effects of SCD inhibition can be at least partially dependent on changes in ratio free MUFA/free SFA. Treatment of cell with inhibited SCD with free MUFA/SFA can serve as useful control for experiments in which different exogenous PI were added to cells.

Answer: Thank you for proposing this valuable control experiment. Following the advice of the reviewer, we treated fibroblasts with CAY10566 in presence or absence of either 18:1 or 16:0 for 48 h. Supplementation of 18:1 restored MUFA-PI and PI(18:1/18:1) levels (new Supplementary Fig. 21a) and prevented i) p38 MAPK activation, ii) the induction of the UPR, and iii) initiation of autophagy, as well as iv) PARP cleavage (new Fig. 21b) in line with our findings for PI(18:1/18:1). Supplementation of 16:0 instead potentiated the cytotoxic activity of CAY10566 (new Supplementary Fig. 21c), which left not sufficient (attached) cells for further mechanistic studies. We now explicitly mention on page 21, line- 26-36 that changes in the ratio of free MUFAs/SFAs (or their metabolites) besides PI(18:1/18:1) likely add to the stress-reductive activity of SCD1, at least for UPR induction in CAY10566-treated fibroblasts, where 18:1 (new Supplementary Fig. 21b) was more efficient than PI(18:1/18:1) (Fig. 6b) in suppressing BiP expression. In addition, we performed quantitative proteomics studies on fibroblasts treated with cytotoxic stressors, CAY10566 or CAY10566 plus PI(18:1/18:1), which provided further insights into the stress(-adaptive) mechanisms of SCD1-derived PI(18:1/18:1) (new Fig. 8 and Supplementary Fig. 23, page 24, line 14 to page 26, line 18, Supplementary

Note 7).

Comment: 6. In support of their hypothesis, authors studied level of PI(18:1/18:1) in different cells, tissues, and organisms being in various physiological states characterized by different stress tolerance such as tumorigenesis, drug-resistance, starvation, aging, and infection. They assess just the level of PI(18:1/18:1) and find both increase and decrease in the amount of this lipid. It looks like without assessment of other players, such as activity of p38, level of UPR and autophagy these results can provide only circumstantial evidence of the role of PI (18:1/18:1) if any.

Answer: While these studies were designed to give a first impression under which physiological and pathophysiological conditions PI(18:1/18:1) levels are regulated, we agree with the reviewer that more in-depth insights into stress-adaptive signaling in exactly these models would be a strong asset. We therefore selected two model systems and confirmed i) autophagy regulation in B-cell lymphoma from E μ -Myc-transgenic mice (new Fig. 9b, page 28, lines 3-11) and ii) p38 MAPK, autophagy-, and apoptosis-related gene expression in starved planarians with silenced Smed-cct3A (new Fig. 9h, Supplementary Table 1, page 30, line 17 to page 31, line 2, Supplementary Note 8).

REVIEWER #2

Comment: The manuscript of Thuermer et al. discusses PI(18:1/18:1) as stress lipokine and biomarker. General remarks: this is a very extensive manuscript and I have to advice condensing it.

Answer: Following the advice of the reviewer, we condensed the manuscript and moved the former Figures 1a, 5, 9c, and 9d to the Supplementary Information. Moreover, we transferred more detailed and specialized descriptions, explanations and discussions to the Supplementary Notes 1 and 3-11, which allowed us to incorporate a substantial amount of novel data into the manuscript (i.e., Fig. 3b, c, Fig. 4l, Fig. 5e, Fig. 8a-c, Fig. 9b, h, Supplementary Fig. 3b, c, Supplementary Fig. 7, Supplementary Fig. 8a, b, Supplementary Fig. 10d, Supplementary Fig. 20b, c, Supplementary Fig. 21a-c, Supplementary Fig. 23a-e, Supplementary Fig. 24a-g, Supplementary Fig. 27a-d, Supplementary Table 1, and Supplementary Data 1-6), as recommended by the reviewers.

Comment: A paper by Pein et al is cited at multiple instances as in preparation. This is inappropriate as I cannot access this information.

Answer: The manuscript by Pein et al. has been uploaded during online submission to be made accessible to the reviewers.

Comment: Indeed SCD1 does produce 18:1-CoA and further as the authors correctly outline "PI(18:1/18:1) is biosynthesized from SCD1-derived 18:1-CoA". However, in my opinion there is a missing puzzle piece here, PIS Phosphatidylinositol synthase. If I am not mistaken is this enzyme ultimately catalyzing the production of PI18:1. However, it seems that it has been left out of the investigation.

Answer: Done, we now analyzed the mRNA expression of phosphatidylinositol synthase (PIS) in CAY10566-treated fibroblasts and found that the terminal enzyme in PI biosynthesis is not significantly regulated (new Supplementary Fig. 24g). Note that PIS is not selective for MUFA-containing CDP- diacylglycerol, and its inhibition can poorly explain changes in the PI fatty acid composition but is rather expected to decrease total PI biosynthesis [PMID: 9370331]. CAY10566 accordingly failed to reduce the absolute amount of PI (new Supplementary Fig. 8b), and cytotoxic stressors evoked diverse results, ranging from a moderate increase to decrease of PI levels (Fig. 1b), which rather excludes PIS as major site of regulation. Moreover, we

performed quantitative proteomics studies on valinomycin- and

myrtoCommulone A-treated fibroblasts to study the effect on PI biosynthesis, subcellular PI distribution, and phosphoinositide metabolism. In this context, we also investigated whether effects are mimicked by SCD1 inhibition (CAY10566) and diminished by additional supplementation of PI(18:1/18:1) (page 23, line 14 to page 24, line 13, new Supplementary Fig. 24, new Supplementary Note 6).

Comment: While I agree that PI18:1 might be a promising stress biomarker is the presented mechanistic evidence on how this exactly works rather descriptive in my view. We know now that SCD/PI18:1/p-MAPK (etc...) are connected but how exactly? ... In other words, this manuscript clearly establishes a link between SCD/PI18:1/MAPK but in my view comes short in explaining how these three are now exactly connected. Lots of interesting observations are presented underlining PI18:1 as possibly useful biomarker, however the causal connections are not sufficiently explored.

Answer: We agree that the mechanistic links between the SCD1-dependent biosynthesis of PI(18:1/18:1) and stress responses (p38 MAPK signaling, the UPR, autophagy, apoptosis) are of high interest. The role of the MAP2Ks MKK-3/6 in activating p38 MAPK upon SCD1 inhibition has already been shown in the previous version of the manuscript (Supplementary Fig. 10b). To identify putative MAP3K that activate MKK-3/6, we additionally analyzed the phosphorylation of ASK1, MLK3, TAK1, and TAO2 using phosphoselective antibodies. Specific bands were obtained for p-MLK3 and p-TAK1, with p-MLK3 being strongly upregulated at 48 h (new Supplementary Fig. 10d and Fig. X4).

Fig. X4. SCD1 inhibition does not affect cellular TAK1 phosphorylation. Fibroblasts were treated with CAY10566 (CAY, 3 μ M) for 48 h and phosphorylation of TAK1 at T187 was determined using a phosphoselective antibody (#ab192443, Abcam). Mean \pm s.e.m. from $n = 4$ independent experiments; two-tailed paired student t -test.

Next, we performed a target fishing approach, for which we i) loaded DEAE- or chitosan-coated magnetic iron oxide particles with PI(18:1/18:1) or the saturated control PI(16:0/16:0), ii) confirmed successful loading by differential light scattering, iii) treated the nanoparticles with fibroblast lysates, iv) recovered the nanoparticles and separated the attached proteins by SDS-PAGE (Fig. X5a), and v) identified the pulled-down proteins using a proteomics approach. Since we did not observe visible bands of different intensity for PI(18:1/18:1)- and PI(16:0/16:0)-treated fibroblasts after Coomassie staining, we divided the gel into 13 sections and subjected all visible bands to semi-quantitative proteomic analysis. Fig. X5 lists the proteins (score > 40) that were only detected in the pull-down fractions of cell homogenates incubated with either PI(18:1/18:1)-coated chitosan (Fig. X5b) or PI(18:1/18:1)-coated DEAE dextran nanoparticles (Fig. X5c).

b

PI(18:1/18:1)-coated chitosan nanoparticles

protein description	score	average mass [Da]	protein matched peptides	sequence coverage [%]	protein amount [ng on column]
60S acidic ribosomal protein	139.71	34387.6	3	16.72	0.199
Developmentally-regulated GTP-binding protein	59.19	40827.3	3	10.63	0.247
Coiled-coil domain-containing protein 47	51.02	56129.1	2	5.59	0.325
RNA 3'-terminal phosphate cyclase-like protein	46.03	41466.1	3	6.17	0.125
DnaJ homolog subfamily A member 3	40.70	53127.9	3	7.29	0.175

c

PI(18:1/18:1)-coated DEAE dextran nanoparticles

protein description	score	average mass [Da]	protein matched peptides	sequence coverage [%]	protein amount [ng on column]
Putative septum site-determining protein	46.53	31127.1	1	3.19	2.672

Fig. X5 Target fishing approach using PI(18:1/18:1)-coated nanoparticles. Chitosan or DEAE dextran iron oxide nanoparticles were coated with PI(18:1/18:1) or PI(16:0/16:0) and incubated with NH-3T3 fibroblast homogenates. **a** SDS-PAGE and Coomassie staining. 'ctrl' indicates control pull-down experiments performed in parallel on chitosan- or DEAE dextran-coated nanoparticles that were not loaded with phospholipids. **b, c** Semi-quantitative proteomics reveals proteins bound to PI(18:1/18:1)-coated chitosan (b) or DEAE dextran nanoparticles (c).

Three of the identified proteins are of particular interest in terms of stress regulation: 1) The coiled-coil domain-containing protein 47 (CCDC47, calumen) is involved in regulating ER Ca²⁺-homeostasis and protects from ER stress-dependent cell death [PMID: 17204322]. 2) The developmentally-regulated GTP-binding protein (DRG1) plays an important role in microtubule organization [PMID: 28855639], and its expression together with c-Myc and Ras stimulates cell transformation in fibroblasts [PMID: 8649774]. 3) The tumor suppressor protein mitochondrial DnaJ homolog subfamily A member 3, also known as tumorous imaginal disc 1 (TID1), belongs to the DNAJ/Hsp40 protein family, which stimulates the ATPase activity of Hsp70 chaperones during protein folding, degradation, and complex assembly [PMID: 16952052]. Notably, TID1 contributes to NFκB signaling [PMID: 31005254] and induces the translocation of p53 to mitochondria, thereby inhibiting apoptosis [PMID: 19935715]. To verify CCDC47, DRG1, and TID1 as PI(18:1/18:1)-interacting proteins, we used specific antibodies and determined their pull-down by Western blot. DRG1 was against our expectations more abundant in samples from PI(16:0/16:0)- than PI(18:1/18:1)-coated nanoparticles (Fig. X6). CCDC47 was present in all fractions, and TID1 was neither specifically detected in the fractions from target fishing nor in cell homogenates (Fig. X6). Since we could not clearly confirm the results from mass spectrometric proteomics (n = 1), we decided not to include the ambiguous target fishing data into the manuscript.

Fig. X6. Immunological detection of DRG1, CCDC47, and TID1 after target fishing with either phospholipid-coated chitosan or DEAE dextran nanoparticles in homogenates of NIH-3T3 fibroblasts. Bound proteins were analyzed by SDS-PAGE and Western blotting.

We conclude from these experiments that either i) the direct targets of PI(18:1/18:1) are only present at low concentrations, ii) PI(18:1/18:1) rather influences membrane substructures without direct interaction with effector proteins, or iii) PI(18:1/18:1) is converted into metabolites (e.g., phosphoinositides, PIP), which then directly interact with the target. Note that phosphatidylinositol-3-kinases (PI3K) use PI and PIP as substrate [PMID: 30462943], regulate membrane trafficking, autophagy, and the UPR [PMID: 31110302, PMID: 20348926], and potentially participate in p38 MAPK activation [PMID: 31551295]. To explore the impact of SCD1 on the PI3K-dependent formation of PIP, we treated fibroblasts with CAY10566 and analyzed cellular phosphatidylinositol-3,4,5-triphosphate (PIP₃) concentrations. CAY10566 hardly affected PIP₃ levels (new Fig. 4l) but interestingly downregulated PIP-specific phosphatases, partially dependent on PI(18:1/18:1) (new Supplementary Fig. 24). We further performed quantitative proteomics studies on CAY10566-treated fibroblasts and investigated whether PI(18:1/18:1) compensates for the inhibition of SCD1 (Fig. 8a-c). Our data show that cytotoxic stress and SCD1-inhibition decrease the cellular levels of the catalytic subunit of protein phosphatase 2A (Fig. 8a), which dephosphorylates p38 MAPK (PMID: 15569672) and central components in p38 MAPK activation (PMID: 18299321, 16407301), UPR induction (PMID: 32457508) and autophagy regulation (PMID: 26310906). Moreover, the revised manuscript now reports on other enzymes besides SCD1 from PI biosynthesis and metabolism that are directly or indirectly regulated (new Supplementary Fig. 24). We also provide mechanistic insights into the regulation of autophagy by PI(18:1/18:1) in Myc-driven murine B-cell lymphoma (new Fig. 9b) and explore stress-related gene expression that is associated with PI(18:1/18:1) depletion in starved planarians with *smed-cct3A* knockdown (new Fig. 9 h, Supplementary Table 1, Supplementary Note 8).

Comment: Does PI18:1 block expression on mRNA level?

Answer: Yes, we now show that the mRNA expression of BiP is induced by CAY10566 and prevented by co-supplementation of PI(18:1/18:1) (new Supplementary Fig. 20c). Other protein stress markers analyzed by us (i.e., p-p38 MAPK, LC3BI/II, PARP) are not regulated by transcription (new Supplementary Fig. 20c) but phosphorylation (p38 MAPK, Fig. 4a), lipidation (LC3B, Fig. 4g), or proteolytic cleavage (PARP, Fig. 4h).

Comment: From F7 and F8 it seems as if phosphorylation of MAPK is affected, as mentioned below also I am not a proteomics person, but I wonder how quantitative such a finding is based on western plot analysis.

Answer: Western blot results are quantitative when based on fluorimetric detection, as done in our studies. The fluorescence signal is namely proportional to the concentration over a wide concentration range in contrast to signals from detection systems based on horse radish peroxidase or alkaline phosphatase [PMID: 25490604].

Comment: As this is a very extensive manuscript I would like to structure this review along the presented figures. Figure 1: What is the added value of 1a? Network analysis has become fancy, but I sometimes question myself what does it add? Could the heatmaps in 1c be clustered? As in all 18:1 lipids in one cluster than this becomes visually a lot clearer.

Answer: Done, we now clustered 18:1-containing lipids in the heatmap of Fig. 1c, Fig. 5c, and Supplementary Fig. 18c and moved the co-regulated lipid network to the supplementary information (Supplementary Fig. 1a).

Comment: The ultimate point of F1 is that cytotoxic conditions decrease MUFA, this is principally a sound conclusion from the provided data. However, what about the free fatty acid and LPI fractions? This might also be an effect of PLA releasing MUFA.

Answer: Good point, we now exposed fibroblasts to cytotoxic conditions for 48 h and analyzed free fatty acids and LPIs. The cellular proportion of the SCD1-derived MUFAs 16:1 and 18:1 strongly decreased for all cytotoxic settings (new Fig. 3b, c and Supplementary Fig. 3b), as expected from the diminished SCD1 expression (Fig. 3e), whereas LPI species (16:0-LPI, 18:0-LPI, 18:1-LPI) were differentially regulated under the four cytotoxic settings we investigated (new Supplementary Fig. 3c). Thus, valinomycin did not affect cellular LPI ratios, whereas thapsigargin, serum depletion, and potentially myrtoicommulone A increased the proportion of distinct species, though without preference for 18:1-LPI, the PLA₂ cleavage product of PI(18:1/18:1). Based on these data, we rather exclude the hydrolysis of PI by PLA₂ isoenzymes as major mechanism behind the drop in PI(18:1/18:1), as now stated on page 12, first paragraph. Our quantitative proteomics study is in line with these findings: cytosolic phospholipase A₂ (with preference for PUFA-containing phospholipids) is non-significantly enriched in stressed fibroblasts and the expression of phospholipase C isoenzymes is rather decreased than elevated (new Supplementary Fig. 24). On the other hand, we found that the availability of LPLAT6/LCLAT1 is reduced under cytotoxic conditions, which might add to the decline of the MUFA-PI content (new Supplementary Fig. 24). LPLAT6 incorporates 18:1 along with other fatty acids into PI (PMID: 34890643), and its levels are maintained by SCD1 in a PI(18:1/18:1)-dependent manner (new Supplementary Fig. 24), as now discussed on page 23, line 18 to page 24, line 6.

Comment: Figure 2: The objective of this paragraph is to correlate PI18:1 with MAPK. Indeed, for some of the tested cell lines there is a correlation between MAPK and PI18:1.

Answer: Thank you for this positive evaluation.

Comment: Figure 3:A – this should be presented as volcano plot. This paragraph shows that there is a correlation between SCD and MUFA/PUFA/SFA fractions. The conclusions here seem sound.

Answer: Done, we replaced the PCA by a volcano plot for valinomycin and myrtoicommulone A (Fig. 3c). Volcano plots for the other cytotoxic conditions are shown together with the PCA in the new Supplementary Fig. 3a, b.

Comment: Figure 4: Panel 4d is misleading, up to now the authors showed MUFA-PI changes as a fraction of total, now suddenly the panel focuses on a specific ratio. How come? Why is this suddenly introduced? The siRNA experiment is crucial. These inhibitors do a thousand things plus inhibiting SCD so I would not be surprised if CAY10566 induces cell death/stress to a certain extend.

Answer: Done, we now replaced in Fig. 4d the ratio PI(18:1/18:1)/PI(18:0/20:4) and PC(18:1/18:1)/PC(16:0/18:1) by the cellular proportion of PI(18:1/18:1) and PC(18:1/18:1), as presented for CAY10566-treated cells. In the previous version of the manuscript, PI(18:1/18:1) and PC(18:1/18:1) levels have been normalized to abundant “house-keeping” phospholipid species to circumvent the need of analyzing the whole set of PI and PC species.

Comment: Along these lines, did CAY10566 also change other lipid fractions? In my view panel E is the most relevant as it shows a correlation between MAPK and SCD knock down.

Answer: We previously published that CAY10566 preferentially decreases MUFA-PI and less MUFA-PS and MUFA-PE levels in NIH-3T3 cells under our experimental conditions (see Fig. 1A in PMID 25678624). However, we did not show the consequences on individual phospholipid species, as now done in Supplementary Fig. 7.

Comment: Figure 5: I am not a proteomics person, but 1-2 matched peptides are in my view not enough for a solid protein identification. Hence, the data in this panel looks quite shaky to me. Please comment.

Answer: Thank you for sharing your concerns. We now transferred the former Figure 5 to the supplementary information (Supplementary Fig. 12). Since peptides get lost during sample preparation and extraction from the gel matrix, it is actually not uncommon in the proteomics field that low abundant proteins are identified for target fishing based on only a few peptides. In addition, peptides will remain unmatched by database searching when posttranslationally modified. To improve the quality of protein identification, we used stringent criteria for database search, and in this study, only accepted confident matches, as described on page 53, line 21 to page 54, line 2.

Comment: Moreover, a PLA response would be quite logic under stress conditions, this would correlate with panel 4i where CAY shows a clear trend of inducing apoptosis, please comment.

Answer: In fact, SCD-1 inhibition has previously been reported to trigger cell death programs including apoptosis [PMID: 27550503] and ferroptosis [PMID: 31270077], for which also a role of phospholipases A₂ has been defined [PMID: 16962822, 25838312, 33542532]. Moreover, distinct cytotoxic stressors used by us (in particular thapsigargin) are reported to activate phospholipases A₂ isoenzymes by rising intracellular Ca²⁺ levels [PMID: 15978132, 10666300]. On the other hand, our novel data strongly suggest that phospholipases A₂ do not drive the cellular depletion of PI(18:1/18:1) in fibroblasts under cytotoxic conditions, as now discussed on page 12, first paragraph. Please also refer to our answer to comment 8.

Comment: Figure 6: Subsequently the authors seek to confirm their in vitro findings in an in vivo model of SCD deficient mice. The mice showed the expected decreased MUFA levels as well as increased MAPK levels etc. Panel 6E I would show as volcano plot.

Answer: Done, we now use Volcano plots in Fig. 5e to compare the PI and PC fatty acid distribution of wildtype and Scd1^{ab-2J} mutant mice for liver, fat, muscle, and skin. The PCA has been moved to the supplementary information (Supplementary Fig. 18e).

Comment: Figure 7: As a next step the authors carry out an “add-back” experiment using PI liposomes. How relevant are the employed 50microM concentrations? How does this relate to the in vitro/in vivo concentration of PI18:1? The rest of this paragraph is clear.

Answer: Thank you for raising this relevant point, which we now discuss on page 20, lines 1-3. In healthy humans after overnight fasting, PI(36:2) and PI(34:2) (i.e., PI(18:1/18:1), PI(16:1/18:1) and the respective isobars) reach plasma concentrations of 4.3 and 2.6 μM, respectively [PMID: 20671299]. PI(18:1/18:1) levels increase after food intake. Thus, rats with ad libitum access to food have average PI(18:1/18:1) plasma concentrations of 16 μM, which further raise to 25 μM at high-fat diet, and the sum of PI(18:1/18:1) and PI(16:1_18:1) reaches mean concentration ranging from 30 to 45 μM [PMID: 31956159]. We consider PI(18:1/18:1) supplementation at 50 μM therefore as a physiological intervention. However, we also want to point out that the direct uptake of phospholipids is not the primary route for cellular phospholipid delivery. Cells more efficiently take up free fatty acids and lysophospholipids, which are then intracellularly used for the re-synthesis of phospholipids via the Kennedy pathway [PMID: 34890643]. To confirm the relevance of this delivery route, we performed compensation studies using non-esterified 18:1 instead of PI(18:1/18:1). Free 18:1 was efficiently incorporated into PI and other phospholipids (new Supplementary Fig. 21a) and diminished p38 MAPK activation, UPR induction, autophagy regulation, and apoptosis-related PARP cleavage comparable to PI(18:1/18:1) (Supplementary Fig. 21b). Note that the here applied 18:1 concentrations of 100 μM are close to the average 18:1 levels (80 μM) in human plasma of fastened individuals [PMID: 20671299]. Moreover, we want to emphasize that supplementation of PI(18:1/18:1) evokes physiological changes of the fibroblast phospholipid composition. Thus, SCD1 inhibition decreased the cellular proportion of PI(18:1/18:1) by a factor of 2.6 (new Supplementary Fig. 8a), whereas supplementation of

PI(18:1/18:1) resulted in a 2- to 4-fold accumulation relative to vehicle control (Supplementary Fig. 20a) independent from SCD1 inhibition (new Supplementary Fig. 20b).

Comment: Figure 8: How do the authors explain the effects of CAY here? In other words, PI18:1 is a SCD derived lipokine which limits stress response (MAPK, URP etc). Now, we add PI18:1 and it blocks p-p38MAPK etc. But when we add PI18:1 together with CAY the effects are absent? Maybe I misunderstand the setup, but CAY mainly blocks SCD which produces PI18:1. However, in this setup we add PI18:1 so SCD is not needed to produce PI18:1. In turn, how does CAY diminish the effects of PI18:1 in this setup?

Answer: Thank you for pointing out this weakness in labeling the bar charts. The label 'CAY' refers to the third and fourth bar and was located directly under the labels 'w/o' or 'PI(18:1/18:1)'. The confusion originates from the 45° orientation of the labels because the bar indicating the 'CAY'- treatment groups could be easily misinterpreted as indicator of the first two bars. We now formatted the labels in Fig. 6, Fig. 7, Supplementary Fig. 20, and Supplementary Fig. 21 in an unambiguous way.

Comment: Figure 9: This paragraph is clear, the authors investigate PI18:1 as possible stress biomarker in several systems.

Answer: We highly appreciate the positive feedback.

Comment: Figure 10: This figure summarizes the findings. As I mentioned above, in my view PIS is missing here and has not been mentioned throughout the manuscript.

Answer: Thank you for bringing PIS to our attention. As outlined in our answer to comment 3 and underlined by new experimental data (Supplementary Fig. 24g), changes in PIS expression seem not to contribute to the drop in PI(18:1/18:1) upon SCD1 inhibition or cytotoxic stress. Based on this outcome, we suggest not to include PIS into the conclusion figure but keep the focus on SCD1.

REVIEWER #3

Comment: The manuscript by Thuermer et al. "PI (18:1/18:1) is a SCD-1 derived lipokine that limits stress signaling presents the data that PI (18:1/18:1), an SCD-1 generated lipid inhibiting p38 MAPK phosphorylation has a strong inhibitory effect on the stress response. The authors thoroughly characterized the effects of the SCD1 inhibition on lipid composition and on stress signaling in various contexts, however, they do not reach deeper into the molecular mechanism explaining the action of PI (18:1/18:1). The topic is appropriate and should garner attention of the broad audience. Aside from the lack of mechanistic explanation there are a few issues that need attention should the manuscript be considered for publication.

Answer: Many thanks for this constructive assessment. We fully agree with the reviewer that a deeper mechanistic understanding of how PI(18:1/18:1) regulates stress signaling would further increase the impact of the manuscript. For this reason, we performed additional experimental studies and investigated the kinase cascade leading to p38 MAPK phosphorylation (new Supplementary Fig. 10d and Fig. X4 in the response to reviewer 2), performed a targeted fishing approach on PI(18:1/18:1)- coated nanoparticles along with unbiased semi-quantitative proteomics (see Figs. X5 and X6 in the response to reviewer 2), determined the consequences of SCD1 inhibition on phosphoinositide levels (new Fig. 4I), and performed quantitative proteomics studies on fibroblasts treated with either CAY10566 or selected cytotoxic stressors, thereby investigating whether PI(18:1/18:1) compensates for the inhibition of SCD1 (new Fig. 8, Supplementary Fig. 23). Please refer to our answer to comment 4 of reviewer 2, where we outline these approaches in more detail. Moreover, the revised manuscript

now proposes concrete mechanisms on how SCD1-derived PI(18:1/18:1) transmits the signal to stress responses (new Fig. 8a-c, Supplementary Note 7, and discussion on page 35, line 10 to page 36, line

17) and reports on additional enzymes in PI biosynthesis, subcellular PI distribution, and phosphoinositide metabolism that are directly or indirectly regulated (new Supplementary Fig. 24). We also provide mechanistic insights into the regulation of autophagy by PI(18:1/18:1) in Myc-driven murine B-cell lymphoma (new Fig. 9b) and explore stress-related gene expression that is associated with PI(18:1/18:1) depletion in starved planarians with *smcd-cct3A* knockdown (new Fig. 9h).

Comment: Detailed comments: 1. The effect of SCD1 inhibition on protein tyrosine phosphorylation. The data in figure 5 are not very convincing. Why did the authors quantify the band showing on Western blot around 100kDa and indeed different between CAY and W/o samples, and then decided to look at two bands of completely different mass that were different on the Coomassie stained gel (which obviously doesn't show only tyrosine phosphorylation but total protein) and present tyrosine phosphorylated peptides which are NOT the dominant protein in these bands? Were the bands close to 100kDa also analyzed? If anything, the whole samples should be analyzed for differential tyrosine phosphorylation using pY-IP. Then these results would be meaningful.

Answer: Agreed, the former panel A was not well connected to the rest of Fig. 5 and has therefore been deleted. Moreover, we transferred Fig. 5 to the supplementary information (Supplementary Fig. 12). The reason for this confusion was that we first investigated tyrosine phosphorylation of proteins from CAY10566-treated samples and found the band close to 100 kDa being strongly elevated. However, when we then immunoprecipitated phospho-tyrosine proteins and stained total proteins with Coomassie, we did not observe a substantial band at 100 kDa and also the faint lines did not suggest an enrichment in the CAY10566-treated samples. Under these conditions, our experimental proteomics setup will hardly be able to unambiguously identify the apparently low-abundant protein, and we therefore decided not to continue this subproject. On the other hand, we identified in the immunoprecipitate further Coomassie-stained bands that are more intense in samples from CAY10566- as compared to vehicle-treated fibroblasts and therefore focused the proteomic target identification exclusively on these more abundant proteins.

Comment: Also, a table showing the processed data should be added as a supplemental table (aside from the raw data submitted to the database).

Answer: Done, we now provide the processed proteomics data as Supplementary Data 1.

Comment: 2. Why was only PI(18:1/18:1) chosen for further investigation? PI(16:0/16:0) had even stronger effect on the BiP levels, this should be at least discussed.

Answer: There seems to be a misunderstanding. Fig. 6b shows that CAY10566-induced BiP expression is efficiently reduced by supplementation of PI(18:1/18:1), whereas PI(16:0/16:0) is hardly active. We improved the labeling of the lanes and describe this finding now more precisely on page 19, lines 17- 19.

Comment: 3. (minor) some persistent spelling errors should be fixed, for example, principal, not principle components (pages 11, 18) and Coomassie, not coomassie (figure 5 legend).

Answer: Done, we carefully revised the manuscript and corrected spelling errors.

Reviewers' Comments:

Reviewer #1:

Remarks to the Author:

Authors performed good job revising the manuscript. They answered to all my concerns and I think, that manuscript can be accepted for publication.

Reviewer #2:

Remarks to the Author:

The authors have taken my comments very serious and I am convinced by their response. This manuscript has become an extensive study underlining a role of PI(18:1) as stress limiting lipokine it will be interesting to see in how far this compound can be leveraged as physiological response marker. I do not have additional comments.

Reviewer #3:

Remarks to the Author:

The authors have very carefully and thoroughly addressed all the comments so in my view the manuscript can be now accepted for publication.

Response to reviewers

REVIEWER #1

Comment: Authors performed good job revising the manuscript. They answered to all my concerns and I think, that manuscript can be accepted for publication.

Answer: Thank you for your kind evaluation of our work.

REVIEWER #2

Comment: The authors have taken my comments very serious and I am convinced by their response. This manuscript has become an extensive study underlining a role of PI(18:1) as stress limiting lipokine it will be interesting to see in how far this compound can be leveraged as physiological response marker. I do not have additional comments.

Answer: Many thanks for this positive and motivating feedback.

REVIEWER #3

Comment: The authors have very carefully and thoroughly addressed all the comments so in my view the manuscript can be now accepted for publication.

Answer: Many thanks, your evaluation is highly appreciated.